# EFFECTIVE DIFFUSION-FREE SCORE MATCHING FOR EXACT CONDITIONAL SAMPLING

## ABSTRACT

The success of score-based models largely stems from the idea of denoising a diffusion process given by a collection of time-indexed score fields. While diffusion-based models have achieved impressive results in sample generation, leveraging them for sound probabilistic inference—particularly for sampling from *arbitrary conditionals of the approximate data distribution*—remains challenging. Briefly, this difficulty arises because conditioning information is only observed for clean data and not available for higher noise levels, which would be required for generating exact conditional samples. In this paper, we introduce an effective approach to *DIffusion-free SCOre matching* (DISCO), which sidesteps the need for time-dependent score fields altogether. Our method is based on a principled objective that, while reminiscent of diffusion-based training, estimates only the score of the (slightly perturbed) data distribution. In our experiments, score models learned with DISCO are competitive with state-of-the-art diffusion models in terms of sample quality. More importantly, DISCO yields a more faithful representation of the underlying data distribution and—crucially—enables accurate sampling from arbitrary conditional distributions, outperforming standard heuristics samplers. This capability opens the door to sound and flexible probabilistic reasoning with score-based models.

## 1 INTRODUCTION

Generative modeling via *score matching* learns the score function rather than the density (Hyvärinen & Dayan, 2005), avoiding the intractable normalization constant. A classical connection to autoencoders leads to effective learning via *denoising score matching* (Vincent, 2011), which, however, fits the score only close to the data manifold, effectively ignoring low-density regions.[1] This fundamental limitation, that leads to brittle sampling routines, has been addressed by generative modeling through reversing a *diffusion process* (Sohl-Dickstein et al., 2015; Song et al., 2020), ensuring the model is fit on a large support, achieving unprecedented sample quality in *diffusion-based score models*.

Yet this success in sample generation masks a critical limitation: by augmenting the *single* data distribution with a *family of noisy copies*, diffusion models fundamentally struggle as probabilistic reasoners. While they excel at producing visually compelling samples, probability theory is really a rigorous framework for reasoning under uncertainty (Jaynes, 1995; Pearl, 1988). Computing *marginals* and *conditionals* are fundamental operations in probabilistic reasoning (Ghahramani, 2015), lying at the core of Bayesian methods, inverse problems and optimal decision making. This raises the pivotal question: **Can we develop score-based models that serve as sound probabilistic reasoners, providing access to *exact* marginals and conditionals?** Here, we focus on drawing faithful *samples* from arbitrary marginals or conditionals—a capability essential for Monte Carlo-based inference and principled uncertainty quantification.

For marginals, the answer is straightforward: one can draw samples from the joint distribution and discard the marginalized variables. Sampling from conditionals, however, exposes the *fundamental inadequacy of diffusion models for probabilistic reasoning*: exact conditional sampling would require conditioning the *entire* diffusion process on the available observations, which is intractable.

---

[1]This is even true when using an actually normalized density as score model, as score matching does not directly incentivize to "pull probability mass towards the data".

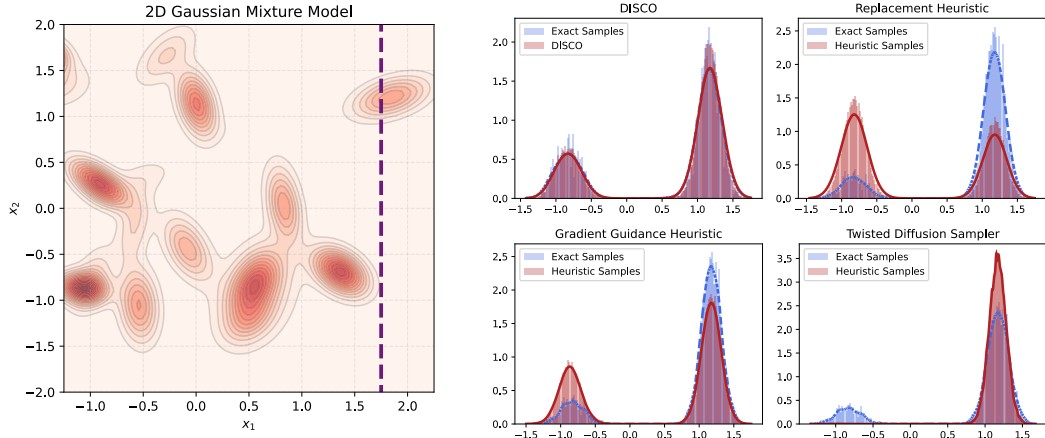

Figure 1: Conditional sampling in a low-dimensional setting. We train an energy-based diffusion model and DISCO model on samples of a 2-dimensional Gaussian mixture model ($p_d$, left). We produce conditional samples from the learned models, $x_2 \sim p_\theta(x_2 \mid x_1 = 1.75)$, and compare these with ground truth conditional samples derived via rejection sampling (right). For the diffusion model we use *Gradient Guidance* (Ho et al., 2022), the *Replacement Heuristic* (Song et al., 2020) and the *Twisted Diffusion Sampler* (TDS) (Wu et al., 2023), for which the produced samples follow a substantially different distribution than the ground truth. In contrast, conditional samples from DISCO (this paper) using tempered SMC follow the ground truth distribution faithfully.

Existing approaches are either ad-hoc heuristics (Song et al., 2020; Ho et al., 2022; Kawar et al., 2022) or provide only asymptotic guarantees (Wu et al., 2023). As demonstrated in Figure 1, *these methods fail to produce unbiased conditional samples—even on elementary toy problems*, revealing the brittleness of diffusion-based probabilistic reasoning.

We introduce *DIffusion-free SCOre matching* (DISCO), which *addresses the probabilistic reasoning problem* by eliminating the need for diffusion-based training altogether. By starting from a mixture of generalized Fisher divergences, specified by an array of "noisy" proposal distributions, we arrive at a principled score matching objective. This objective, albeit reminiscent to diffusion training, only fits the (slightly perturbed) data distribution rather than a full diffusion process, while taking care that the score field is also fit outside the data manifold. This approach makes conditioning de-facto trivial: one simply fixes observed variables in the learned score field and samples only the unobserved variables—enabling asymptotically exact probabilistic inference.

Our experiments demonstrate that DISCO not only matches state-of-the-art diffusion models in sample quality—achieving competitive FID scores on FFHQ-64 and CIFAR-10—but *dramatically outperforms them in probabilistic fidelity*. DISCO provides a faithful representation of the underlying data distribution and delivers accurate conditional sampling across both low- and high-dimensional problems, where diffusion-based methods systematically struggle. *This enables principled probabilistic reasoning with score-based models*, unlocking their potential for rigorous uncertainty quantification, Bayesian inference, and scientific modeling.

## 2 BACKGROUND

### 2.1 SCORE-BASED MODELING AND SAMPLING

In generative modeling, we are given i.i.d. samples $\{\mathbf{x}^{(i)} \in \mathbb{R}^D\}_{i=1}^N$ from a data distribution $p_d(\mathbf{x})$, and aim to learn a parametric model $p_\theta$ that approximates $p_d$ well. Score-based modeling (Hyvärinen & Dayan, 2005) circumvents the challenge of normalization by learning the *score* of the data density, defined as $\nabla_\mathbf{x} \log p_d(\mathbf{x})$, which is invariant to the normalizing constant. The idea is to use a neural network $s_\theta : \mathbb{R}^D \to \mathbb{R}^D$ to represent the model score and minimize the *Fisher divergence*:

$$\mathcal{F}(p_d \,\|\, s_\theta) := \mathbb{E}_{\mathbf{x} \sim p_d} \left[ \|\nabla_\mathbf{x} \log p_d(\mathbf{x}) - s_\theta(\mathbf{x})\|_2^2 \right] \tag{1}$$

Alternatively, one can define a score model via an *energy-based model*, giving rise to a proper density $p_{\boldsymbol{\theta}}(\mathbf{x}) := \exp(-E_{\boldsymbol{\theta}}(\mathbf{x}))/Z_{\boldsymbol{\theta}}$, where the score $\nabla_{\mathbf{x}} \log p_{\boldsymbol{\theta}}(\mathbf{x}) = -\nabla_{\mathbf{x}} E_{\boldsymbol{\theta}}(\mathbf{x})$ can be computed using automatic differentiation. Energy-based models define a valid distribution and enable asymptotically unbiased sampling via Monte Carlo methods such as Hamiltonian Monte Carlo (HMC) (Neal et al., 2011).

In this paper, we consider both unconstrained score fields and energy-based models. However, since our focus is on probabilistic reasoning, energy-based models are preferable, as they define a valid distribution and enable asymptotically unbiased sampling via Monte Carlo methods.

## 2.2 DENOISING SCORE MATCHING

Since the Fisher divergence in (1) involves the unknown score $\nabla_{\mathbf{x}} \log p_d(\mathbf{x})$, it is generally unsuitable for direct optimization. This motivates the use of alternative objectives that do not require explicit access to $\nabla_{\mathbf{x}} \log p_d(\mathbf{x})$. A particularly popular variant is *denoising score matching* (DSM), which approximates the score of a perturbed data distribution $p_\sigma(\tilde{\mathbf{x}}) = \int p_d(\mathbf{x}) \, q(\tilde{\mathbf{x}} \,|\, \mathbf{x}) \, d\mathbf{x}$, where $q(\tilde{\mathbf{x}} \,|\, \mathbf{x}) := \mathcal{N}(\tilde{\mathbf{x}} \,|\, \mathbf{x}, \sigma^2 I)$ is a Gaussian perturbation kernel with *fixed* noise level $\sigma$. Concretely, minimizing the objective

$$\mathcal{L}_{\text{DSM}}(\boldsymbol{\theta}) := \mathbb{E}_{\mathbf{x} \sim p_d(\mathbf{x}), \tilde{\mathbf{x}} \sim q(\tilde{\mathbf{x}} \,|\, \mathbf{x})} \left[ \|\nabla_{\tilde{\mathbf{x}}} \log q(\tilde{\mathbf{x}} \,|\, \mathbf{x}) - s_{\boldsymbol{\theta}}(\tilde{\mathbf{x}})\|_2^2 \right] \tag{2}$$

is equivalent to minimizing Fisher divergence, as $\nabla_{\boldsymbol{\theta}} \mathcal{L}_{\text{DSM}}(\boldsymbol{\theta}) = \nabla_{\boldsymbol{\theta}} \mathcal{F}(p_\sigma \,\|\, s_{\boldsymbol{\theta}})$ for all $\boldsymbol{\theta}$ (Vincent, 2011). This objective and its gradients can be efficiently estimated using data samples, as it only depends on the score of the perturbation kernel, given by $\nabla_{\tilde{\mathbf{x}}} \log q(\tilde{\mathbf{x}} \,|\, \mathbf{x}) = (\mathbf{x} - \tilde{\mathbf{x}})/\sigma^2$.

In (2), one chooses the fixed noise level $\sigma$ to be small so that the perturbed distribution $p_\sigma$ closely approximates the data distribution $p_d$. However, this implies that the score is mainly learned near the data manifold, even though $p_\sigma$ formally has full support on $\mathbb{R}^D$. In regions far from the manifold, $p_\sigma$ almost never samples points, so the learned score is essentially arbitrary there. Since sampling methods are generally initialized far from the data manifold, inaccurate score estimates in these low-density regions cause the sampler to drift toward arbitrary directions, producing poor samples (Song & Ermon, 2019).

## 2.3 DIFFUSION MODELS

Diffusion models address the limitations of naïve DSM by learning a *multitude* of score vector fields, each corresponding to a different noise level applied to the data distribution (Sohl-Dickstein et al., 2015; Song et al., 2020). Formally, let the clean data be denoted by $\mathbf{x}_0 \sim p_d$, and define the conditional distribution $p_t(\mathbf{x}_t \,|\, \mathbf{x}_0)$ via the forward diffusion process:

$$\mathbf{x}_t = \alpha(t)\mathbf{x}_0 + \sigma(t)\boldsymbol{\varepsilon}, \qquad \boldsymbol{\varepsilon} \sim \mathcal{N}(\mathbf{0}, I) \tag{3}$$

where $t \in [0, T]$ for some $T > 0$. In this work, we focus primarily on the variance-exploding (VE) formulation (Song et al., 2020), where $\alpha(t) = 1$ and only the noise scale $\sigma(t)$ varies over time. This process defines a family of progressively noisier distributions $\{p_t(\mathbf{x}_t)\}_{t \in [0,T]}$, where $p_t(\mathbf{x}_t) = \int q_t(\mathbf{x}_t \,|\, \mathbf{x}_0) \, p_d(\mathbf{x}_0) \, d\mathbf{x}_0$ and $q_t(\mathbf{x}_t \,|\, \mathbf{x}_0) = \mathcal{N}(\mathbf{x}_t \,|\, \mathbf{x}_0, \sigma(t)^2 I)$.

A *time-dependent* score network is then trained to approximate the score function $s_{\boldsymbol{\theta}}(\mathbf{x}, t) \approx \nabla_{\mathbf{x}} \log p_t(\mathbf{x})$ for all $\mathbf{x} \in \mathbb{R}^D$ and $t \in [0, T]$, by minimizing

$$\mathcal{L}_{\text{DM}}(\boldsymbol{\theta}) = \mathbb{E}_{t, \mathbf{x}_0, \mathbf{x}_t} \left[ \lambda(t) \|\nabla_{\mathbf{x}_t} \log p_t(\mathbf{x}_t \,|\, \mathbf{x}_0) - s_{\boldsymbol{\theta}}(\mathbf{x}_t, t)\|_2^2 \right], \tag{4}$$

where $t \sim p(t)$, $\mathbf{x}_0 \sim p_d(\mathbf{x}_0)$, and $\mathbf{x}_t \sim q_t(\mathbf{x}_t \,|\, \mathbf{x}_0)$. Here $p(t)$ is some distribution over $[0, T]$ and $\lambda(t)$ is a positive weighting function. Note that (4) is basically an extension of (2) to a time-indexed family of score fields $\{\nabla_{\mathbf{x}} \log p_t(\mathbf{x}_t)\}_{t \in [0,T]}$, approximated by a shared score network $s_{\boldsymbol{\theta}}(\mathbf{x}_t, t)$.

After training, the score network $s_{\boldsymbol{\theta}}$ is used for sample generation, aiming to approximate draws from $p_0$. Popular approaches are numerical integration of the reverse-time SDE (Song et al., 2020) and ancestral sampling (Ho et al., 2020). A key advantage of diffusion models over standard DSM is that, due to training across multiple noise levels, the score network is also informed in low-density regions. Empirically, this leads to high-quality samples and has established diffusion models as the current state of the art in generative modeling.

## 3 DIFFUSION-FREE SCORE MATCHING

Although diffusion models enable high-quality sample generation, they introduce the overhead of an entire family of score functions, where only the approximate data score at $t = 0$ is of actual interest. Even though $\{\mathbf{x}_t\}_{t>0}$ are merely "noisy copies" of $\mathbf{x}_0 = \mathbf{x}$, they are strictly speaking *latent variables* which make *conditional sampling*—a fundamental operation for probabilistic inference (Ghahramani, 2015)—highly challenging. Specifically, when splitting the data variable $\mathbf{x}$ into **u**nobserved variables $\mathbf{x}^u$ and **c**onditioned variables $\mathbf{x}^c$, the goal is to sample $\mathbf{x}^u \sim p(\mathbf{x}^u \,|\, \mathbf{x}^c)$. When dealing with only a *single* score field $\nabla_{\mathbf{x}} \log p(\mathbf{x})$, conditioning becomes straightforward, since the conditional score is simply the joint score with clamped $\mathbf{x}^c$:

$$\nabla_{\mathbf{x}^u} \log p(\mathbf{x}^u, \mathbf{x}^c) = \nabla_{\mathbf{x}^u} \log p(\mathbf{x}^u \,|\, \mathbf{x}^c) + \overbrace{\nabla_{\mathbf{x}^u} \log p(\mathbf{x}^c)}^{=\mathbf{0}} \qquad (5)$$

However, drawing conditional samples with diffusion models requires $\nabla_{\mathbf{x}_t} \log p_t(\mathbf{x}_t \,|\, \mathbf{x}_0^c)$ for each $t > 0$, which is intractable to compute. While much work has been devoted to derive conditional samples from diffusion models (Song et al., 2020; Ho et al., 2022; Kawar et al., 2022; Wu et al., 2023), this task remains challenging.

In this paper, we reconsider the assumption that diffusion-based learning is strictly necessary for learning expressive score-based models. Instead, we aim to learn just a *single* score field, which allows us to sample any conditional according to (5). To this end, we start with a slight modification of the Fisher divergence:

**Definition 1.** *q-Weighted Fisher Divergence. Let $p_d$ and $q$ be probability densities over $\mathbb{R}^D$ whose supports satisfy* $\mathrm{supp}(p_d) \subseteq \mathrm{supp}(q)$. *We define the $q$-weighted Fisher divergence as*

$$\mathcal{F}_q(p_d \,\|\, s_{\boldsymbol{\theta}}) := \mathbb{E}_{\mathbf{x} \sim q} \left[ \|\nabla_{\mathbf{x}} \log p_d(\mathbf{x}) - s_{\boldsymbol{\theta}}(\mathbf{x})\|_2^2 \right]. \qquad (6)$$

Like the Fisher divergence $\mathcal{F}$ in Equation (1), also $\mathcal{F}_q$ measures the score-mismatch between $p_d$ and the model $s_{\boldsymbol{\theta}}$, but in expectation over a *proposal distribution* $q$ rather than $p_d$. It is easy to show that $\mathcal{F}_q(p_d \,\|\, s_{\boldsymbol{\theta}}) = 0$ implies $\mathcal{F}(p_d \,\|\, s_{\boldsymbol{\theta}}) = 0$, hence $\mathcal{F}_q$ is a principled divergence.

Next, we adopt from diffusion models the idea of using a family of Gaussian perturbed distributions where

$$q_t(\mathbf{x}_t \,|\, \mathbf{x}) := \mathcal{N}(\mathbf{x}_t \,|\, \mathbf{x}, \sigma(t)^2 I) \qquad \text{is a Gaussian perturbation kernel indexed by } t \in [0, T] \qquad (7)$$

$$p_t(\mathbf{x}_t, \mathbf{x}) = q_t(\mathbf{x}_t \,|\, \mathbf{x}) \, p_d(\mathbf{x}) \qquad \text{is the joint of a data sample } \mathbf{x} \text{ and a perturbed version } \mathbf{x}_t \qquad (8)$$

$$p_t(\mathbf{x}_t) = \int p_t(\mathbf{x}_t, \mathbf{x}) \, \mathrm{d}\mathbf{x} \qquad \text{is the marginal distribution of } \mathbf{x}_t \text{ derived from (8)} \qquad (9)$$

Below we will further need the *posterior* over the data sample $\mathbf{x}$ conditional on a perturbed version $\mathbf{x}_t$, given as

$$p_{t'}(\mathbf{x} \,|\, \mathbf{x}_t) = \frac{p_{t'}(\mathbf{x}_t, \mathbf{x})}{p_{t'}(\mathbf{x}_t)}. \qquad (10)$$

Note that we adopted the parameter $t$ from diffusion models, which in our case does not signify time but just indexes noise levels $\sigma(t)$, monotonously increasing with $t$. We further introduce, similar as in diffusion models, a prior distribution $p(t)$ over $t \in [0, T]$ and a continuous positive weighting function $\lambda(t)$.

Unlike as in diffusion models, we do not aim to approximate the $p_t(\mathbf{x}_t)$'s for $t > 0$, but use them merely as proposals for $\mathcal{F}_q$. We propose to minimize a *weighted mixture of q-weighted Fisher divergences*:

$$\mathcal{F}_{\mathrm{mix}}(p_d \,\|\, s_{\boldsymbol{\theta}}) = \mathbb{E}_{t \sim p(t)} \left[ \lambda(t) \, \mathcal{F}_{p_t}(p_d \,\|\, s_{\boldsymbol{\theta}}) \right] \qquad (11)$$

$$= \mathbb{E}_{t \sim p(t)} \left[ \lambda(t) \, \mathbb{E}_{\mathbf{x}_t \sim p_t} \left[ \|\nabla_{\mathbf{x}_t} \log p_d(\mathbf{x}_t) - s_{\boldsymbol{\theta}}(\mathbf{x}_t)\|_2^2 \right] \right]. \qquad (12)$$

Also $\mathcal{F}_{\mathrm{mix}}$ is a principled objective, since, as $\lambda(t)$ is positive and $\mathcal{F}_{p_t}$ is non-negative, $\mathcal{F}_{\mathrm{mix}}(p_d \,\|\, s_{\boldsymbol{\theta}}) = 0$ implies that $\mathcal{F}_{p_t}(p_d \,\|\, s_{\boldsymbol{\theta}}) = 0$ for almost all $t \in [0, T]$.

$\mathcal{F}_{\mathrm{mix}}$ requires the true data score $\nabla_{\mathbf{x}} \log p_d(\mathbf{x})$ which is not available. Hence, we adopt a similar approach as in (Vincent, 2011) and replace $p_d$ with a slightly Gaussian-perturbed version $p_d'(\mathbf{x}) := p_0(\mathbf{x})$, i.e. the perturbed data distribution (9) at the lowest noise level. Given that $\sigma(0)$ is small, fitting $p_d'$ instead of $p_d$ is a worthwhile goal. With this modification, we are able to derive the following principled objective, the *DIffusion-free SCOre matching* loss (DISCO loss):

**Theorem 1.** *Let $p_d$ be the true data distribution, $p(t)$ a distribution over $[0, T]$, and $\lambda(t)$ a positive weighting function. Further, let $q_t(\mathbf{x}' \mid \mathbf{x})$, $p_t(\mathbf{x}_t)$ and $p_t(\mathbf{x} \mid \mathbf{x}_t)$ be defined as in (7), (9), and (10), respectively. Let $q(t, \mathbf{x}, \mathbf{x}_t) := p_0(\mathbf{x} \mid \mathbf{x}_t)\, p_t(\mathbf{x}_t)\, p(t)$. The DISCO loss*

$$\mathcal{L}_{\text{DISCO}}(\boldsymbol{\theta}) := \mathbb{E}_{q(t, \mathbf{x}, \mathbf{x}_t)} \left[ \lambda(t) \left\| \nabla_{\mathbf{x}_t} \log q_0(\mathbf{x}_t \mid \mathbf{x}) - s_{\boldsymbol{\theta}}(\mathbf{x}_t) \right\|_2^2 \right] \tag{13}$$

*has the same parameter gradients as $\mathcal{F}_{mix}(p'_d \parallel s_{\boldsymbol{\theta}})$.*

The proof can be found in Appendix A.1. From Theorem 1 it follows that, given that $s_{\boldsymbol{\theta}}$ has sufficient capacity, the global minimizer of $\mathcal{L}_{\text{DISCO}}$ will learn the true score of $p'_d$. In (13) it can be seen that $\mathcal{L}_{\text{DISCO}}$ shares features from both $\mathcal{L}_{\text{DSM}}$ (2) and $\mathcal{L}_{\text{DM}}$ (4): Like in DSM, the DISCO objective only fits a single score model corresponding to a slightly perturbed data distribution. In particular, note that $s_{\boldsymbol{\theta}}$ does not need to depend on $t$. Furthermore, similar as in (2), we only require the score of the (smallest) perturbation kernel, given as $\nabla_{\mathbf{x}_t} \log q_0(\mathbf{x}_t \mid \mathbf{x}) = (\mathbf{x} - \mathbf{x}_t)/\sigma(0)^2$. Like in diffusion models, (13) computes a weighted expectation over various noisy data distributions, making sure that $s_{\boldsymbol{\theta}}$ gets informed far from the data manifold. Crucially, comparing the DISCO loss to diffusion training, we re-interpret the diffused distributions $p_t$ for $t > 0$ merely as proposal distributions in $q$-weighted Fisher divergences and do *not* learn their score fields.

**DISCO Training.** Estimating $\mathcal{L}_{\text{DISCO}}$ for training is straightforward, except for one part. In order to sample from $q(t, \mathbf{x}, \mathbf{x}_t)$, we first sample $t \sim p(t)$. Subsequently, we sample $\mathbf{x}_t \sim p_t(\mathbf{x}_t)$, by first sampling some (intermediate) data sample $\mathbf{x}' \sim p_d$ and then its perturbed version $\mathbf{x}_t \sim q_t(\mathbf{x}_t \mid \mathbf{x}')$. The challenging part is then to sample $p_0(\mathbf{x} \mid \mathbf{x}_t)$.[2] However, as we usually have only finitely many training data points $\mathcal{D} = \{\mathbf{x}^{(i)}\}_{i=1}^N$, the data distribution is the empirical distribution $p_d(\mathbf{x}) = p_{\text{emp}}(\mathbf{x}) := \frac{1}{N} \sum_{i=1}^N \delta(\mathbf{x}^{(i)} - \mathbf{x})$ where $\delta(\cdot)$ denotes the Dirac-delta function. From Bayes' law, we obtain

$$p_0(\mathbf{x} \mid \mathbf{x}_t) = \frac{q_0(\mathbf{x}_t \mid \mathbf{x})\, p_{\text{emp}}(\mathbf{x})}{p_0(\mathbf{x}_t)}$$

which induces a probability mass function over $\mathcal{D}$. Thus, we compute $p_0(\mathbf{x}^{(i)} \mid \mathbf{x}_t) \propto q_0(\mathbf{x}_t \mid \mathbf{x}^{(i)})$ for each $\mathbf{x}^{(i)} \in \mathcal{D}$ and sample $\mathbf{x}$ from the normalized mass function. If $|\mathcal{D}|$ is large, we can draw an approximate posterior sample by either (1) constructing the probability mass function using a mini-batch of data $\mathcal{B} \subset \mathcal{D}$, or (2) using special data structures that allow finding the $k$ nearest neighbors of $\mathbf{x}_t$ quickly, which can be used to approximate $p_0(\mathbf{x} \mid \mathbf{x}_t)$. A more detailed discussion can be found in Appendix B.

**Masked DISCO Training.** During conditional sampling, the model must have learned the score at points $(\mathbf{x}^u, \mathbf{x}^c)$ where $\mathbf{x}^c$ is "clean", and $\mathbf{x}^u$ is "noisy". While in theory, minimizing $\mathcal{L}_{\text{DISCO}}$ learns the true score also at these points, we observe that in high dimensions, the model does not learn accurate scores at these points. Therefore, we introduce a variant of the DISCO loss that keeps the global minimum unchanged, but facilitates learning the score at these points. Specifically, consider a distribution $p(\mathbf{m})$ over binary masks $\mathbf{m} \in \{0, 1\}^D$ and let $\mathbf{x}_{t, \mathbf{m}}$ and $\mathbf{x}_{t, \bar{\mathbf{m}}}$ denote the coordinates of $\mathbf{x}_t$ where the mask is 1 and 0, respectively. Let $P_{\bar{\mathbf{m}}}$ denote the projection on the coordinates where $\mathbf{m}$ is 0 and let $q_t$ again denote the Gaussian perturbation kernel. With $p_t(\mathbf{x}_{t, \bar{\mathbf{m}}}, \mathbf{x}_{\mathbf{m}}) = \int q_t(\mathbf{x}_{t, \bar{\mathbf{m}}} \mid \mathbf{x}_{\bar{\mathbf{m}}})\, p_d(\mathbf{x}_{\bar{\mathbf{m}}}, \mathbf{x}_{\mathbf{m}})\, d\mathbf{x}_{\bar{\mathbf{m}}}$, we define $q(t, \mathbf{m}, \mathbf{x}_{t, \bar{\mathbf{m}}}, \mathbf{x}_{\mathbf{m}}, \mathbf{x}_{\bar{\mathbf{m}}}) = p(t)p(\mathbf{m})p_t(\mathbf{x}_{t, \bar{\mathbf{m}}}, \mathbf{x}_{\mathbf{m}})p_0(\mathbf{x}_{\bar{\mathbf{m}}} \mid \mathbf{x}_{t, \bar{\mathbf{m}}}, \mathbf{x}_{\mathbf{m}})$ and

$$\mathcal{L}_{\text{DISCO}}^{\text{mask}}(\boldsymbol{\theta}) := \mathcal{L}_{\text{DISCO}}(\boldsymbol{\theta}) + \gamma L_{\text{mask}}^{\text{DISCO}}(\boldsymbol{\theta}) \quad \text{with} \tag{14}$$

$$L_{\text{mask}}^{\text{DISCO}}(\boldsymbol{\theta}) := \mathbb{E}_{q(t, \mathbf{m}, \mathbf{x}_{t, \bar{\mathbf{m}}}, \mathbf{x}_{\mathbf{m}}, \mathbf{x}_{\bar{\mathbf{m}}})} \left[ \lambda(t) \left\| \nabla_{\mathbf{x}_{t, \bar{\mathbf{m}}}} \log q_0(\mathbf{x}_{t, \bar{\mathbf{m}}} \mid \mathbf{x}_{\bar{\mathbf{m}}}) - P_{\bar{\mathbf{m}}} s_{\theta}(\mathbf{x}_{t, \bar{\mathbf{m}}}, \mathbf{x}_{\mathbf{m}}) \right\|_2^2 \right] \tag{15}$$

where $\gamma > 0$. In other words, we put Gaussian noise only on a subset of variables and only care about the network's predictions for these noisy variables. Whenever $\mathbf{m} \neq \mathbf{0}$, sampling from the posterior $p_0(\mathbf{x}_{\bar{\mathbf{m}}} \mid \mathbf{x}_{t, \bar{\mathbf{m}}}, \mathbf{x}_{\mathbf{m}})$ amounts to just taking the previously sampled $\mathbf{x}_{\bar{\mathbf{m}}}$. We prove the fact that the global minimum of $\mathcal{L}_{\text{DISCO}}^{\text{mask}}$ is the same as that of $\mathcal{L}_{\text{DISCO}}$ in Appendix A.3.

---

[2]Note the asymmetry in this principle, where $\mathbf{x}_t$ is generated by a perturbation at "high" noise levels, but the posterior $p_0(\mathbf{x} \mid \mathbf{x}_t)$ is over clean data "assuming $\mathbf{x}_t$ had been generated by $p_0$ (lowest noise level)." In particular, the intermediate sample $\mathbf{x}'$ which was used to produce $\mathbf{x}_t$ does *not* necessarily have high probability under $p_0(\mathbf{x} \mid \mathbf{x}_t)$, especially for large $\sigma(t)$.

**Masked Diffusion Model Training.** To provide a baseline for the masked DISCO training, we analogously define a masked diffusion model training objective:

$$\mathcal{L}_{\mathrm{DM}}^{\mathrm{mask}}(\boldsymbol{\theta}) := \mathcal{L}_{\mathrm{DM}}(\boldsymbol{\theta}) + \gamma L_{\mathrm{mask}}^{\mathrm{DM}}(\boldsymbol{\theta}) \quad \text{with} \tag{16}$$

$$L_{\mathrm{mask}}^{\mathrm{DM}}(\boldsymbol{\theta}; \mathbf{m}) = \mathbb{E}_{p(t)p(\mathbf{m})p_d(\mathbf{x})q_t(\mathbf{x}_{t,\bar{\mathbf{m}}}|\mathbf{x}_{\bar{\mathbf{m}}})} \left[ \left\| \nabla_{\mathbf{x}_{t,\bar{\mathbf{m}}}} \log q_t(\mathbf{x}_{t,\bar{\mathbf{m}}} \mid \mathbf{x}_{\bar{\mathbf{m}}}) - P_{\bar{\mathbf{m}}} s_{\boldsymbol{\theta}}(\mathbf{x}_t, t, \mathbf{m}) \right\|_2^2 \right] \tag{17}$$

where $\mathbf{x}_t = (\mathbf{x}_{t,\bar{\mathbf{m}}}, \mathbf{x}_{\mathbf{m}})$ and $\gamma > 0$. For the empty mask $\mathbf{m} = \mathbf{0}$, the global optimum of $\mathcal{L}_{\mathrm{DM}}^{\mathrm{mask}}$ is $s_{\boldsymbol{\theta}}^*(\mathbf{x}_t, t, \mathbf{m} = \mathbf{0}) = \nabla_{\mathbf{x}_t} \log p_t(\mathbf{x}_t)$. For other $\mathbf{m}$, the global optimum is $P_{\bar{\mathbf{m}}} s_{\boldsymbol{\theta}}^*(\mathbf{x}_t, t, \mathbf{m}) = \nabla_{\mathbf{x}_{t,\bar{\mathbf{m}}}} \log p_t(\mathbf{x}_{t,\bar{\mathbf{m}}} \mid \mathbf{x}_{\mathbf{m}})$. Note that this is merely circumventing the problem of probabilistic inference by directly trying to learn the otherwise intractable quantities. Therefore, we have no guarantee that the conditionals we learn are *consistent*, i.e., that there exists a joint distribution whose conditionals coincide with the learned ones. Clearly, this breaks sound probabilistic reasoning, and hence, these models should also be considered a heuristic baseline, rather than a principled approach.

## 4 RELATED WORK

**Time-Independence in Score-Based Models.** Most similar in spirit to DISCO is the work by Li et al. (Li et al., 2023), who share the idea of only learning $\nabla_{\mathbf{x}} \log p_0(\mathbf{x})$ using a score-matching objective. However, they do not minimize $\mathcal{L}_{\mathrm{DISCO}}$, but a variant which they term *multiscale denoising score matching (MDSM)*, which is $\mathcal{L}_{\mathrm{DISCO}}$ when (incorrectly) setting $q(t, \mathbf{x}, \mathbf{x}_t) := p(t)p_d(\mathbf{x})p_t(\mathbf{x}_t \mid \mathbf{x})$ in (13). This objective in fact learns $s_{\boldsymbol{\theta}}^*(\mathbf{x}_t) = \mathbb{E}_{p(t \mid \mathbf{x}_t)} \left[ \frac{\sigma(t)^2}{\sigma(0)^2} \nabla_{\mathbf{x}_t} \log p_t(\mathbf{x}_t) \right]$, i.e. a *posterior average over $p_t$ scores*, where the posterior over noise levels is reweighted. Thus, the claim of (Li et al., 2023) that $s_{\boldsymbol{\theta}}^*$ only learns the score of $p_0$ is erroneous (see Appendix A.2). Their main motivation is also not conditional sampling but on analyzing diffusion training.

A key property in DISCO is that the score network is independent of $t$, while diffusion-based models inherently rely on a notion of time. Yet, there have been attempts to minimize $\mathcal{L}_{\mathrm{DM}}$ with neural networks where (1) time enters in a simple way, or (2) time does not enter into the network $s_{\boldsymbol{\theta}}(\mathbf{x})$ at all. Song & Ermon (2020) proposed to model $s_{\boldsymbol{\theta}}(\mathbf{x}, t) := \varepsilon_{\boldsymbol{\theta}}(\mathbf{x})/\sigma(t)$ where $\varepsilon_{\boldsymbol{\theta}}$ is a time-independent neural network. However, it is easy to see that the *true* scores of different noise levels are *not* just scaled versions of another, i.e., there exists no constant $c$ such that $\nabla_{\mathbf{x}} \log p_{t_1}(\mathbf{x}) = c \cdot \nabla_{\mathbf{x}} \log p_{t_2}(\mathbf{x}) \ \forall \mathbf{x}, t_1 \neq t_2$, except for the trivial case where $p_0$ is Gaussian. Thus, even with infinite capacity in $\varepsilon_{\boldsymbol{\theta}}$, we cannot learn the true scores. In fact, one can interpret this parameterization as learning a single distribution whose *tempered* versions try to match the diffused distributions $p_t$. Recently, Sun et al. (2025) studied the effect of minimizing $\mathcal{L}_{\mathrm{DM}}$ with a time-independent network $s_{\boldsymbol{\theta}}(\mathbf{x})$. Doing so results in a minimizer $s_{\boldsymbol{\theta}}^*(\mathbf{x}_t) = \mathbb{E}_{p(t \mid \mathbf{x}_t)} [\nabla_{\mathbf{x}_t} \log p_t(\mathbf{x}_t)]$, which learns to average the scores of $p_t$ over the posterior distribution of noise levels (see Appendix A.4). Sun et al. (2025) argue that in high dimensions, $p(t \mid \mathbf{x}_t)$ is close to $\delta(t - t_{\mathbf{x}_t})$, where $\mathbf{x}_t = \mathbf{x}_0 + t_{\mathbf{x}_t} \varepsilon, \varepsilon \sim \mathcal{N}(0, I)$, and hence, $s_{\boldsymbol{\theta}}^*(\mathbf{x}_t) \approx \nabla_{\mathbf{x}_t} \log p_{t_{\mathbf{x}_t}}(\mathbf{x}_t)$. However, this work is clearly distinct to DISCO, as we try to regress $\nabla_{\mathbf{x}} \log p_0(\mathbf{x})$ only.

**Conditional Sampling in Diffusion Models.** Many approximations to the true conditional $p_0(\mathbf{x}_0^u \mid \mathbf{x}_0^c)$ have been proposed: Song et al. (2020) introduce the *replacement method*, a popular heuristic that estimates the conditional score at time $t$ as

$$\nabla_{\mathbf{x}_t^u} \log p_t(\mathbf{x}_t^u \mid \mathbf{x}_0^c) \approx \nabla_{\mathbf{x}_t^u} \log p_t(\mathbf{x}_t^u \mid \hat{\mathbf{x}}_t^c) \tag{18}$$

where $\hat{\mathbf{x}}_t^c$ is drawn from the known distribution $p_t(\mathbf{x}_t^c \mid \mathbf{x}_0^c) = \mathcal{N}(\mathbf{x}_t^c; \alpha(t)\mathbf{x}_0^c, \sigma(t)^2 I)$. This approximation enjoys no theoretical guarantees and often fails to produce samples coherent with the conditioning information (Ho et al., 2022).

*Gradient guidance* (Ho et al., 2022) relies on the fact that $\nabla_{\mathbf{x}_t} \log p_t(\mathbf{x}_t \mid \mathbf{x}_0^c) = \nabla_{\mathbf{x}_t} \log p_t(\mathbf{x}_0^c \mid \mathbf{x}_t) + \nabla_{\mathbf{x}_t} \log p_t(\mathbf{x}_t)$. While $\nabla_{\mathbf{x}_t} \log p_t(\mathbf{x}_t)$ is known via $s_{\boldsymbol{\theta}}$, the intractable quantity $p_t(\mathbf{x}_0^c \mid \mathbf{x}_t)$ is approximated, often by $\mathcal{N}(\mathbf{x}_0^c; \Omega(\hat{\mathbf{x}}_{\boldsymbol{\theta}}(\mathbf{x}_t, t)), \sigma(t)^2 I)$, where $\hat{\mathbf{x}}_{\boldsymbol{\theta}}(\mathbf{x}, t) = \mathbf{x} + \sigma(t)^2 s_{\boldsymbol{\theta}}(\mathbf{x}, t)$ is the "denoised" input, and $\Omega(\mathbf{x})$ returns only the observed coordinates in $\mathbf{x}$. At each noise level $t$, the approximation of the conditional score $\nabla_{\mathbf{x}_t} \log p_t(\mathbf{x}_t^u \mid \mathbf{x}_t^c)$ is used to perform sampling. Note that computing $\nabla_{\mathbf{x}_t} \log \mathcal{N}(\mathbf{x}_0^c; \Omega(\hat{\mathbf{x}}_{\boldsymbol{\theta}}(\mathbf{x}_t, t)), \sigma(t)^2 I)$ involves backpropagating through the neural network, making this approximation computationally expensive. Again, this heuristic provides unreliable estimates (Zhang et al., 2023) and comes with no theoretical guarantees.

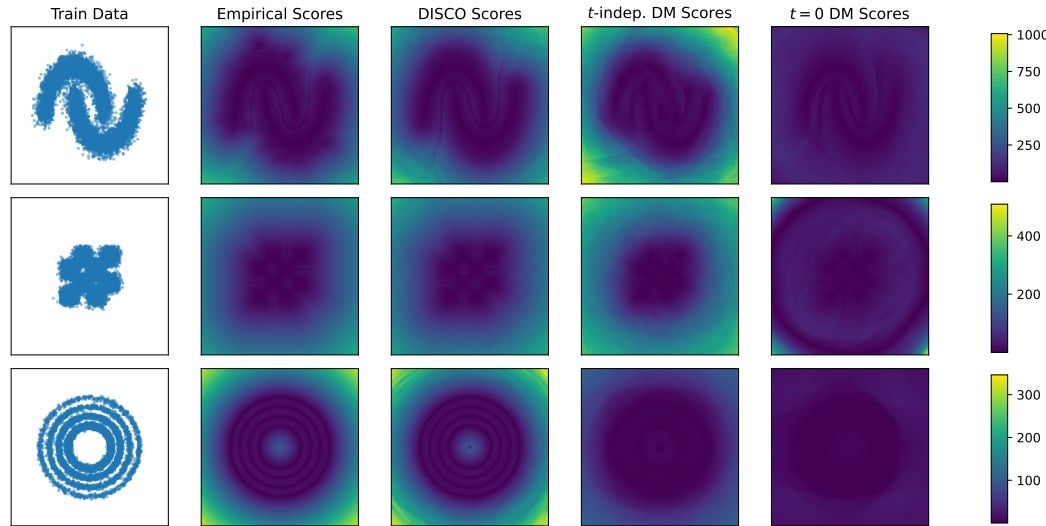

Figure 2: **Qualitative Results on 2D Distributions:** Comparison of the $L_2$ norms of the scores of the ground truth empirical distribution, an energy-based DISCO model, an energy-based diffusion model (DM) that is independent of $t$ (i.e., trained on $\mathcal{L}_{\text{DM}}$ but $E_{\boldsymbol{\theta}}(\mathbf{x})$ has no access to $t$, as in Sun et al. (2025)), and an energy-based DM at $t = 0$. Samples from $p'_d$ are shown on the very left (training data). Only DISCO estimates the magnitudes of the true scores faithfully.

Table 1: **Quantitative Evaluation on 2D Distributions:** *Model Fit (joint)* estimates the Wasserstein-1 distance $W_1\left(p'_d(\mathbf{x}), p_{\boldsymbol{\theta}}(\mathbf{x})\right)$ by sampling from the Diffusion Model via 100 steps of ancestral sampling, and sampling from DISCO using our tempered SMC sampler. *Inference Quality* forms a Monte Carlo estimate of $\mathbb{E}_{p'_d(x_1)}\left[W_1\left(p_{\boldsymbol{\theta}}(x_2 \mid x_1), \hat{p}_{\boldsymbol{\theta}}(x_2 \mid x_1)\right)\right]$, where we sample from the true *model* conditional $p_{\boldsymbol{\theta}}(x_2 \mid x_1)$ via rejection sampling, and we use popular heuristics to draw approximate conditional samples, whose law is denoted as $\hat{p}_{\boldsymbol{\theta}}(x_2 \mid x_1)$. DISCO is on-par or better in terms of model fit, while allowing for almost error-free conditional inference. All shown results are the mean $\pm$ two standard deviations over independent runs.

|  | Method | Moons | Checkerboard | Rings |
|---|---|---|---|---|
| *Model Fit (joint)* | Diffusion Model | **0.062 ± 0.009** | 0.057 ± 0.004 | 0.081 ± 0.003 |
|  | DISCO | 0.068 ± 0.022 | **0.029 ± 0.008** | **0.066 ± 0.026** |
| *Inference Quality* | Replacement | 0.363 ± 0.409 | 0.141 ± 0.129 | 0.099 ± 0.074 |
|  | Grad. Guidance | 0.417 ± 0.487 | 0.165 ± 0.129 | 0.117 ± 0.077 |
|  | TDS | 0.326 ± 1.016 | 0.120 ± 0.165 | 0.082 ± 0.067 |
|  | DISCO | **0.024 ± 0.025** | **0.015 ± 0.012** | **0.042 ± 0.040** |

(Wu et al., 2023) introduced the *twisted diffusion sampler* (TDS), which uses gradient guidance in a twisted sequential Monte Carlo (SMC) procedure as an approximation to the (unknown) optimal twisting function. Due to this, the sampler will not produce exact samples for any *finite* number of simulated particles. In contrast, DISCO guarantees asymptotically exact samples, even when simulating *a single particle*.

## 5 EXPERIMENTS

### 5.1 LOW-DIMENSIONAL SETTING

To experimentally validate DISCO in a low-dimensional setting, we train both a regular energy-based diffusion model and an energy-based DISCO model on various 2D distributions (Gaussian Mixture Model, Moons, Checkerboard, Rings). and compare the quality of samples from the condi-

Figure 3: **Qualitative Inpainting Results:** Comparison of inpainting methods on FFHQ-64. Best viewed zoomed in. More results can be found in Appendix E.

| **FFHQ** | *Wide* | | *Narrow* | | *Super-Resolve 2x* | | *Altern. Lines* | | *Half* | | *Expand* | |
| Method | LPIPS↓ | SSIM↑ | LPIPS↓ | SSIM↑ | LPIPS↓ | SSIM↑ | LPIPS↓ | SSIM↑ | LPIPS↓ | SSIM↑ | LPIPS↓ | SSIM↑ |
|---|---|---|---|---|---|---|---|---|---|---|---|---|
| EDM Masked | **0.104** | **0.797** | **0.016** | **0.960** | **0.048** | **0.880** | **0.022** | **0.942** | **0.158** | 0.692 | **0.433** | **0.242** |
| Replacement | 0.192 | 0.673 | 0.048 | 0.916 | 0.410 | 0.562 | 0.106 | 0.851 | 0.218 | 0.595 | 0.502 | 0.150 |
| Grad. Guidance | 0.140 | 0.749 | 0.027 | 0.941 | 0.114 | 0.797 | 0.037 | 0.914 | 0.178 | 0.665 | 0.481 | 0.186 |
| RePaint | 0.136 | 0.767 | 0.027 | 0.942 | 0.107 | 0.820 | 0.057 | 0.903 | 0.184 | 0.670 | 0.491 | 0.211 |
| TDS | 0.142 | 0.786 | 0.028 | 0.951 | 0.136 | 0.829 | 0.059 | 0.916 | 0.189 | **0.697** | 0.488 | 0.209 |
| DISCO | 0.119 | 0.772 | 0.027 | 0.944 | 0.068 | 0.838 | 0.026 | 0.926 | 0.166 | 0.675 | 0.450 | 0.210 |

| **CIFAR-10** | *Wide* | | *Narrow* | | *Super-Resolve 2x* | | *Altern. Lines* | | *Half* | | *Expand* | |
| Method | LPIPS↓ | SSIM↑ | LPIPS↓ | SSIM↑ | LPIPS↓ | SSIM↑ | LPIPS↓ | SSIM↑ | LPIPS↓ | SSIM↑ | LPIPS↓ | SSIM↑ |
|---|---|---|---|---|---|---|---|---|---|---|---|---|
| EDM Masked | **0.146** | **0.792** | 0.018 | 0.965 | **0.091** | **0.835** | **0.048** | **0.906** | **0.215** | **0.648** | **0.510** | 0.192 |
| Replacement | 0.255 | 0.658 | 0.064 | 0.916 | 0.431 | 0.516 | 0.187 | 0.767 | 0.296 | 0.549 | 0.607 | 0.094 |
| Grad. Guidance | 0.203 | 0.732 | 0.036 | 0.943 | 0.196 | 0.727 | 0.082 | 0.860 | 0.262 | 0.600 | 0.581 | 0.125 |
| RePaint | 0.225 | 0.732 | 0.078 | 0.914 | 0.339 | 0.676 | 0.184 | 0.806 | 0.285 | 0.593 | 0.627 | 0.140 |
| TDS | 0.232 | 0.760 | 0.053 | 0.937 | 0.255 | 0.746 | 0.124 | 0.857 | 0.308 | 0.623 | 0.630 | **0.230** |
| DISCO | 0.163 | 0.789 | **0.018** | **0.970** | 0.119 | 0.832 | 0.052 | 0.902 | 0.231 | 0.626 | 0.541 | 0.161 |

Table 2: **Quantitative Inpainting Results:** On both FFHQ-64 and CIFAR-10, DISCO outperforms all inpainting heuristics that rely on a pre-trained model, and is very close to the best model that was trained on all conditionals, *EDM Masked*, while providing (1) conditional distributions consistent with the learned joint, and (2) better unconditional samples on FFHQ-64 (see Table 3).

tional distribution $p_\theta(x_2 \,|\, x_1)$, using various sampling techniques. Note that in all cases, the baselines correspond to exact conditional samples from the respective *models*, not from the ground truth distribution, as we aim to evaluate each model's ability to generate conditional samples consistent with its own learned joint distribution. These exact samples are obtained via rejection sampling, i.e., by drawing from the joint model $\mathbf{x} \sim p_\theta(x_1, x_2)$ and retaining only those samples that satisfy the condition on $x_1$ (up to a small numerical threshold $\epsilon$).

The qualitative results in Figure 1 demonstrate that conditional samples are faithful only in the DISCO model, while all other methods fail to preserve the relative weights of the Gaussian components. Moreover, Figure 2 shows that only DISCO is able to accurately learn the scores of the true empirical distribution, while the $t = 0$ diffusion model heavily underestimate the score magnitudes, especially far from the training data. This is to be expected, as the diffusion formalism does not even strive to represent a single data score, but "distributes" the generative process over a hierarchy of time-dependent score-fields. To ablate this effect, we also train a time-independent diffusion model (as in (Sun et al., 2025)), which tries to learn a posterior average of $p_t$ scores, and thus again fails to match $\nabla_\mathbf{x} \log p_0(\mathbf{x})$.

Finally, the quantitative results in Table 1 make it clear that our time-independent DISCO model performs just as well as the diffusion models in terms of model fit, while offering asymptotically—and practically—exact conditional samples. More experimental details, including model architectures and sampler hyperparameters, are provided in Appendix C.

## 5.2 IMAGE DATASETS

To demonstrate that DISCO performs well in high-dimensional generative modeling tasks, we train an unconstrained score model with DISCO on both the CIFAR-10 dataset (Krizhevsky et al., 2009)

Table 3: FID scores on unconditional image generation. Qualitative examples can be found in Appendix E.

| Dataset | EDM | EDM Masked | DISCO |
|---------|-----|------------|-------|
| CIFAR-10 | 1.97 | 2.59 | 3.58 |
| FFHQ-64 | 2.39 | 5.71 | 2.65 |

and on FFHQ-64 (Karras et al., 2019). To this end, we use the popular *EDM* implementation (Karras et al., 2022) which defines a *denoising network* $D_{\boldsymbol{\theta}}(\mathbf{x})$, where the score network is then given as

$$s_{\boldsymbol{\theta}}(\mathbf{x}) := \frac{D_{\boldsymbol{\theta}}(\mathbf{x}) - \mathbf{x}}{\sigma(0)^2} \tag{19}$$

Since $\nabla_{\mathbf{x}_t} \log q_0(\mathbf{x}_t \mid \mathbf{x}_0) = (\mathbf{x}_0 - \mathbf{x}_t)/\sigma(0)^2$, it follows that $\mathcal{L}_{\text{DISCO}}$ then simplifies to

$$\mathcal{L}_{\text{DISCO}}(\boldsymbol{\theta}) = \sigma(0)^{-4} \, \mathbb{E}_{q(t,\mathbf{x},\mathbf{x}_t)} \left[ \lambda(t) \| \mathbf{x} - D_{\boldsymbol{\theta}}(\mathbf{x}_t) \|_2^2 \right] \tag{20}$$

where we simply drop $\sigma(0)^{-4}$ because it is a constant factor w.r.t. $\boldsymbol{\theta}$. Karras et al. (2022) model their time-dependent denoiser as

$$D_{\boldsymbol{\theta}}(\mathbf{x}, t) := c_{\text{skip}}(t)\mathbf{x} + c_{\text{out}}(t)F_{\boldsymbol{\theta}}(c_{\text{in}}(t)\mathbf{x}, c_{\text{noise}}(t)) \tag{21}$$

where $F_{\boldsymbol{\theta}}(\cdot, \cdot)$ is the direct output of the neural network, and $c_{\text{skip}}, c_{\text{out}}, c_{\text{in}}, c_{\text{noise}}$ are scalar-valued functions. Since we model a time-independent denoiser, we do not use $c_{\text{noise}}$ and let the network $F_{\boldsymbol{\theta}}$ predict both $c_{\text{skip}}$ and $c_{\text{out}}$ via a linear layer with a sigmoid activation at the output. Finally, we train $D_{\boldsymbol{\theta}}(\mathbf{x})$ by minimizing $\mathcal{L}_{\text{DISCO}}^{\text{mask}}$ with $\gamma = 0.5$, using the mini-batch approximation to sample from the posterior $p_0(\mathbf{x} \mid \mathbf{x}_t)$ (see Section 3). As a baseline, we train *EDM Masked*, which is equivalent to the original *EDM* implementation, except that it minimizes $\mathcal{L}_{\text{DM}}^{\text{mask}}$ with $\gamma = 0.5$ instead of $\mathcal{L}_{\text{DM}}$. Our mask distribution $p(\mathbf{m})$ includes randomly sized patches, and factorized Bernoulli distributions for the mask variables.

We note that we do not tailor our method specifically towards image inpainting—it is merely one application of conditional inference. Therefore, we benchmark against the same general purpose heuristics as in the low-dimensional case, with the addition of *RePaint* (Lugmayr et al., 2022): It uses the replacement heuristic, but introduces "resampling" steps that take an intermediate state $\mathbf{x}_t$, diffuses it back to $\mathbf{x}_{t+1}$ (using the forward process), and denoises it again (Lugmayr et al., 2022) (see Appendix C for more details).

To sample unconditionally and conditionally from DISCO or *EDM Masked*, we use the second-order Heun sampler with 18 steps (i.e., NFE = 35) in all CIFAR-10 experiments, and 40 steps (i.e., NFE = 79) in all FFHQ-64 experiments (Karras et al., 2022). Inpainting results are reported both qualitatively in Figure 3, and quantitatively in Table 2: We evaluate inpainting quality using both the Learned Perceptual Image Patch Similarity (LPIPS) (Zhang et al., 2018) and the Structural Similarity Index Measure (SSIM) (Wang et al., 2004). Unconditional sample quality is reported in Table 3, which demonstrates that directly learning a *single* score vector field can lead to high visual sample quality.

## 6 CONCLUSION

In this paper, we challenge the prevailing belief that diffusion processes are essential for training effective score-based generative models. We introduce DISCO, a diffusion-free score matching framework that avoids time-indexed score fields in favor of learning a single, time-independent score function. Our results demonstrate that this approach is not only viable but also competitive with diffusion models in terms of visual sample quality. More importantly, DISCO provides a principled foundation for exact conditional sampling, which has remained elusive for traditional diffusion-based models. This ability opens the door to using such models as sound probabilistic reasoners, positioning DISCO as powerful tool for a wide array of tasks in probabilistic modeling, beyond mere sample generation. For example, our method might be beneficial for designing molecular structures that satisfy target binding affinities or for sampling physically plausible protein conformations conditioned on partial structural constraints.

**Ethical Statement.** DISCO provides a principled and practical alternative to diffusion-based training for generative modeling, with the added benefit of enabling exact conditional sampling. This has the potential to broaden the applicability of score-based models in domains where precise probabilistic reasoning is critical—such as scientific discovery, computational biology, and decision-making under uncertainty. At the same time, as with other generative models, there is a risk of misuse in the automated generation of misleading or harmful content. However, given that DISCO emphasizes controllable and interpretable sampling over raw visual fidelity, its potential for direct misuse in areas like misinformation or content generation appears limited. Still, any deployment of DISCO-based models should consider the reliability of conditional constraints, dataset biases, and domain-specific safety requirements—especially in sensitive fields like healthcare or public policy.

**Reproducibility Statement.** Upon acceptance, we will make both our code and trained models publicly available, such that all experiments can be reproduced with little effort. Moreover, we have detailed the experimental setup both in the main text and in the appendix (see Section C).

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

# A PROOFS

For notational convenience, we will refer to the Gaussian perturbation kernel $q_t(\mathbf{x}_t \mid \mathbf{x})$ as $p_t(\mathbf{x}_t \mid \mathbf{x})$ throughout the appendix.

## A.1 DISCO OBJECTIVE

Let $p(t)$ be a prior distribution over a "time" parameter[3] $t \in [0, T]$, let $p_d$ denote the data distribution, and let $\alpha : [0, T] \rightarrow \mathbb{R}_{>0}$ and $\sigma : [0, T] \rightarrow \mathbb{R}_{>0}$ be positive functions of time. Given the distributions $p(t, \mathbf{x}, \mathbf{x}_t) := p(t)p_d(\mathbf{x})p_t(\mathbf{x}_t \mid \mathbf{x})$ with $p_t(\mathbf{x}_t \mid \mathbf{x}) := \mathcal{N}(\mathbf{x}_t; \alpha(t)\mathbf{x}, \sigma(t)^2 I)$ and $q(t, \mathbf{x}, \mathbf{x}_t) := p(t)p_t(\mathbf{x}_t)p_0(\mathbf{x} \mid \mathbf{x}_t)$ with $p_t(\mathbf{x}_t) = \int p_t(\mathbf{x}_t \mid \mathbf{x})p_d(\mathbf{x}) \, d\mathbf{x}$ and

$$p_0(\mathbf{x} \mid \mathbf{x}_t) = \frac{p_0(\mathbf{x}_t \mid \mathbf{x})p_d(\mathbf{x})}{p_0(\mathbf{x}_t)},$$

we will show that the *DISCO Loss*

$$\mathcal{L}_{\mathrm{DISCO}}(\boldsymbol{\theta}) := \mathbb{E}_{q(t,\mathbf{x},\mathbf{x}_t)} \left[ \lambda(t) \| \nabla_{\mathbf{x}_t} \log p_0(\mathbf{x}_t \mid \mathbf{x}) - s_{\boldsymbol{\theta}}(\mathbf{x}_t) \|_2^2 \right] \tag{22}$$

is equivalent to

$$\mathcal{F}_{\mathrm{mix}}(p_0 \, \| \, s_{\boldsymbol{\theta}}) = \mathbb{E}_{p(t)} \left[ \lambda(t) \mathcal{F}_{p_t}(p_0 \, \| \, s_{\boldsymbol{\theta}}) \right] = \mathbb{E}_{p(t)} \mathbb{E}_{p_t(\mathbf{x}_t)} \left[ \lambda(t) \| \nabla_{\mathbf{x}_t} \log p_0(\mathbf{x}_t) - s_{\boldsymbol{\theta}}(\mathbf{x}_t) \|_2^2 \right] \tag{23}$$

up to an additive constant independent of $\boldsymbol{\theta}$. As defined above, $p_0(\mathbf{x})$ is the slightly Gaussian-perturbed version of $p_d$ and is also called $p_d'$ in the main text.

*Proof.* We see that

$$\begin{aligned}
\mathcal{L}_{\mathrm{DISCO}}(\boldsymbol{\theta}) &= \mathbb{E}_{q(t,\mathbf{x},\mathbf{x}_t)} \left[ \lambda(t) \| \nabla_{\mathbf{x}_t} \log p_0(\mathbf{x}_t \mid \mathbf{x}) - s_{\boldsymbol{\theta}}(\mathbf{x}_t) \|_2^2 \right] \\
&= \mathbb{E}_{p(t)p_t(\mathbf{x}_t)} \left[ \lambda(t) \, \mathbb{E}_{p_0(\mathbf{x} \mid \mathbf{x}_t)} \left[ \| \nabla_{\mathbf{x}_t} \log p_0(\mathbf{x}_t \mid \mathbf{x}) - s_{\boldsymbol{\theta}}(\mathbf{x}_t) \|_2^2 \right] \right]
\end{aligned}$$

We have

$$\begin{aligned}
& \mathbb{E}_{p_0(\mathbf{x} \mid \mathbf{x}_t)} \left[ \| \nabla_{\mathbf{x}_t} \log p_0(\mathbf{x}_t \mid \mathbf{x}) - s_{\boldsymbol{\theta}}(\mathbf{x}_t) \|_2^2 \right] \\
&= \mathbb{E}_{p_0(\mathbf{x} \mid \mathbf{x}_t)} \left[ \| \nabla_{\mathbf{x}_t} \log p_0(\mathbf{x}_t \mid \mathbf{x}) \|_2^2 - 2 \nabla_{\mathbf{x}_t} \log p_0(\mathbf{x}_t \mid \mathbf{x})^\top s_{\boldsymbol{\theta}}(\mathbf{x}_t) + \| s_{\boldsymbol{\theta}}(\mathbf{x}_t) \|_2^2 \right] \\
&= c_1 - 2 \mathbb{E}_{p_0(\mathbf{x} \mid \mathbf{x}_t)} \left[ \nabla_{\mathbf{x}_t} \log p_0(\mathbf{x}_t \mid \mathbf{x}) \right]^\top s_{\boldsymbol{\theta}}(\mathbf{x}_t) + \| s_{\boldsymbol{\theta}}(\mathbf{x}_t) \|_2^2 \\
&= c_2 + \| \mathbb{E}_{p_0(\mathbf{x} \mid \mathbf{x}_t)} \left[ \nabla_{\mathbf{x}_t} \log p_0(\mathbf{x}_t \mid \mathbf{x}) \right] - s_{\boldsymbol{\theta}}(\mathbf{x}_t) \|_2^2
\end{aligned}$$

where $c_1, c_2$ are constants w.r.t. $\boldsymbol{\theta}$. We notice that

$$\begin{aligned}
\mathbb{E}_{p_0(\mathbf{x} \mid \mathbf{x}_t)} \left[ \nabla_{\mathbf{x}_t} \log p_0(\mathbf{x}_t \mid \mathbf{x}) \right] &= \int p_0(\mathbf{x} \mid \mathbf{x}_t) \nabla_{\mathbf{x}_t} \log p_0(\mathbf{x}_t \mid \mathbf{x}) \, d\mathbf{x} \\
&= \int p_0(\mathbf{x} \mid \mathbf{x}_t) \frac{\nabla_{\mathbf{x}_t} p_0(\mathbf{x}_t \mid \mathbf{x})}{p_0(\mathbf{x}_t \mid \mathbf{x})} \, d\mathbf{x} \\
&= \int \frac{p_0(\mathbf{x}_t \mid \mathbf{x})p_0(\mathbf{x})}{p_0(\mathbf{x}_t)} \frac{\nabla_{\mathbf{x}_t} p_0(\mathbf{x}_t \mid \mathbf{x})}{p_0(\mathbf{x}_t \mid \mathbf{x})} \, d\mathbf{x} \\
&= \int \frac{p_0(\mathbf{x}) \nabla_{\mathbf{x}_t} p_0(\mathbf{x}_t \mid \mathbf{x})}{p_0(\mathbf{x}_t)} \, d\mathbf{x} \\
&= \frac{1}{p_0(\mathbf{x}_t)} \int p_0(\mathbf{x}) \nabla_{\mathbf{x}_t} p_0(\mathbf{x}_t \mid \mathbf{x}) \, d\mathbf{x} \\
&= \frac{1}{p_0(\mathbf{x}_t)} \nabla_{\mathbf{x}_t} \int p_0(\mathbf{x}) p_0(\mathbf{x}_t \mid \mathbf{x}) \, d\mathbf{x} \\
&= \frac{1}{p_0(\mathbf{x}_t)} \nabla_{\mathbf{x}_t} p_0(\mathbf{x}_t) \\
&= \nabla_{\mathbf{x}_t} \log p_0(\mathbf{x}_t)
\end{aligned}$$

---

[3]We want to stress that it has only the meaning of time in diffusion models, while in DISCO it indexes a family of successively noisier proposal distributions.

and hence,

$$\|\mathbb{E}_{p_0(\mathbf{x}\,|\,\mathbf{x}_t)}\left[\nabla_{\mathbf{x}_t}\log p_0(\mathbf{x}_t\,|\,\mathbf{x})\right] - s_{\boldsymbol{\theta}}(\mathbf{x}_t)\|_2^2 = \|\nabla_{\mathbf{x}_t}\log p_0(\mathbf{x}_t) - s_{\boldsymbol{\theta}}(\mathbf{x}_t)\|_2^2$$

which implies that

$$\begin{aligned}
\mathcal{L}_{\text{DISCO}}(\boldsymbol{\theta}) &= \mathbb{E}_{p(t)p_t(\mathbf{x}_t)}\left[\lambda(t)\,\mathbb{E}_{p_0(\mathbf{x}\,|\,\mathbf{x}_t)}\left[\|\nabla_{\mathbf{x}_t}\log p_0(\mathbf{x}_t\,|\,\mathbf{x}) - s_{\boldsymbol{\theta}}(\mathbf{x}_t)\|_2^2\right]\right] + \text{const.} \\
&= \mathbb{E}_{p(t)p_t(\mathbf{x}_t)}\left[\lambda(t)\,\|\nabla_{\mathbf{x}_t}\log p_0(\mathbf{x}_t) - s_{\boldsymbol{\theta}}(\mathbf{x}_t)\|_2^2\right] + \text{const.} \\
&= \mathbb{E}_{p(t)}\left[\lambda(t)\mathcal{F}_{p_t}(p_0\,\|\,s_{\boldsymbol{\theta}})\right] + \text{const.}
\end{aligned}$$

which concludes the proof. $\square$

## A.2 MULTISCALE DENOISING SCORE MATCHING

We show that the *multiscale denoising score matching (MDSM)* (Li et al., 2023) objective

$$\mathcal{L}_{\text{MDSM}}(\boldsymbol{\theta}) = \mathbb{E}_{p(t)p_d(\mathbf{x})p_t(\mathbf{x}_t\,|\,\mathbf{x})}\left[\lambda(t)\,\|\nabla_{\mathbf{x}_t}\log p_0(\mathbf{x}_t\,|\,\mathbf{x}) - s_{\boldsymbol{\theta}}(\mathbf{x}_t)\|_2^2\right] \tag{24}$$

has the minimizer $s_{\boldsymbol{\theta}}^*(\mathbf{x}_t) = \mathbb{E}_{p(t\,|\,\mathbf{x}_t)}\left[\frac{\sigma(t)^2}{\sigma(0)^2}\nabla_{\mathbf{x}_t}\log p_t(\mathbf{x}_t)\right]$ when $\lambda(t) = 1$ and $\alpha(t) = 1$ for all $t$ (variance exploding).

*Proof.* For convenience, we assume $\lambda(t) = 1$, as this can always be subsumed into the prior $p(t)$ without affecting the minimizer. Moreover, we assume $\alpha(t) = 1$. With $p(t, \mathbf{x}, \mathbf{x}_t) = p(t)p_d(\mathbf{x})p_t(\mathbf{x}_t\,|\,\mathbf{x})$, we denote with $p(\mathbf{x}_t)$ the marginal over $\mathbf{x}_t$ (not to be confused with $p_t(\mathbf{x}_t)$, which conditions on $t$). We have

$$\begin{aligned}
\mathcal{L}_{\text{MDSM}}(\boldsymbol{\theta}) &= \mathbb{E}_{p(t)p_d(\mathbf{x})p_t(\mathbf{x}_t\,|\,\mathbf{x})}\left[\|\nabla_{\mathbf{x}_t}\log p_0(\mathbf{x}_t\,|\,\mathbf{x}) - s_{\boldsymbol{\theta}}(\mathbf{x}_t)\|_2^2\right] & (25) \\
&= \mathbb{E}_{p(\mathbf{x}_t)p(t\,|\,\mathbf{x}_t)p_t(\mathbf{x}\,|\,\mathbf{x}_t)}\left[\|\nabla_{\mathbf{x}_t}\log p_0(\mathbf{x}_t\,|\,\mathbf{x}) - s_{\boldsymbol{\theta}}(\mathbf{x}_t)\|_2^2\right] & (26) \\
&= \mathbb{E}_{p(\mathbf{x}_t)p(t\,|\,\mathbf{x}_t)}\left[\|\mathbb{E}_{p_t(\mathbf{x}\,|\,\mathbf{x}_t)}\left[\nabla_{\mathbf{x}_t}\log p_0(\mathbf{x}_t\,|\,\mathbf{x})\right] - s_{\boldsymbol{\theta}}(\mathbf{x}_t)\|_2^2\right] + \text{const.} & (27)
\end{aligned}$$

where the last step follows the same argument as in Section A.1. With $R(\mathbf{x}_t, t) := \mathbb{E}_{p_t(\mathbf{x}\,|\,\mathbf{x}_t)}\left[\nabla_{\mathbf{x}_t}\log p_0(\mathbf{x}_t\,|\,\mathbf{x})\right]$ and repeating this argument, we see that

$$\mathbb{E}_{p(\mathbf{x}_t)p(t\,|\,\mathbf{x}_t)}\left[\|R(\mathbf{x}_t, t) - s_{\boldsymbol{\theta}}(\mathbf{x}_t)\|_2^2\right] = \mathbb{E}_{p(\mathbf{x}_t)}\left[\|\mathbb{E}_{p(t\,|\,\mathbf{x}_t)}\left[R(\mathbf{x}_t, t)\right] - s_{\boldsymbol{\theta}}(\mathbf{x}_t)\|_2^2\right] + \text{const.} \tag{28}$$

where clearly, the minimizer is

$$\begin{aligned}
s_{\boldsymbol{\theta}}^*(\mathbf{x}_t) &= \mathbb{E}_{p(t\,|\,\mathbf{x}_t)}\left[R(\mathbf{x}_t, t)\right] & (29) \\
&= \mathbb{E}_{p(t\,|\,\mathbf{x}_t)p_t(\mathbf{x}\,|\,\mathbf{x}_t)}\left[\nabla_{\mathbf{x}_t}\log p_0(\mathbf{x}_t\,|\,\mathbf{x})\right]. & (30)
\end{aligned}$$

Expanding $\nabla_{\mathbf{x}_t}\log p_0(\mathbf{x}_t\,|\,\mathbf{x}) = (\mathbf{x} - \mathbf{x}_t)/\sigma(0)^2$, we get

$$s_{\boldsymbol{\theta}}^*(\mathbf{x}_t) = \mathbb{E}_{p(t\,|\,\mathbf{x}_t)p_t(\mathbf{x}\,|\,\mathbf{x}_t)}\left[\frac{\mathbf{x} - \mathbf{x}_t}{\sigma(0)^2}\right] = \mathbb{E}_{p(t\,|\,\mathbf{x}_t)}\left[\frac{\mathbb{E}_{p_t(\mathbf{x}\,|\,\mathbf{x}_t)}\left[\mathbf{x}\right] - \mathbf{x}_t}{\sigma(0)^2}\right] \tag{31}$$

Via Tweedie's formula, we can express the posterior mean as $\mathbb{E}_{p_t(\mathbf{x}\,|\,\mathbf{x}_t)}\left[\mathbf{x}\right] = \mathbf{x}_t + \sigma(t)^2\nabla_{\mathbf{x}_t}\log p_t(\mathbf{x}_t)$, and thus,

$$s_{\boldsymbol{\theta}}^*(\mathbf{x}_t) = \mathbb{E}_{p(t\,|\,\mathbf{x}_t)}\left[\frac{\mathbf{x}_t + \sigma(t)^2\nabla_{\mathbf{x}_t}\log p_t(\mathbf{x}_t) - \mathbf{x}_t}{\sigma(0)^2}\right] = \mathbb{E}_{p(t\,|\,\mathbf{x}_t)}\left[\frac{\sigma(t)^2}{\sigma(0)^2}\nabla_{\mathbf{x}_t}\log p_t(\mathbf{x}_t)\right] \tag{32}$$

which concludes the proof. $\square$

This shows that the claim made in Li et al. (2023) that $s_{\boldsymbol{\theta}}^*(\mathbf{x})$ only learns $\nabla_{\mathbf{x}}\log p_0(\mathbf{x})$ is incorrect.

## A.3 MASKED DISCO TRAINING

We prove that the global minimum of $\mathcal{L}_{\text{DISCO}}^{\text{mask}}(\boldsymbol{\theta})$ is the same as that of $\mathcal{L}_{\text{DISCO}}(\boldsymbol{\theta})$.

*Proof.* Recall that

$$\mathcal{L}_{\text{DISCO}}^{\text{mask}}(\boldsymbol{\theta}) = \mathcal{L}_{\text{DISCO}}(\boldsymbol{\theta}) + \gamma L_{\text{mask}}^{\text{DISCO}}(\boldsymbol{\theta}) \tag{33}$$

where

$$L_{\text{mask}}^{\text{DISCO}}(\boldsymbol{\theta}) = \mathbb{E}_{q(t,\mathbf{m},\mathbf{x}_{t,\bar{\mathbf{m}}},\mathbf{x}_{\mathbf{m}},\mathbf{x}_{\bar{\mathbf{m}}})} \left[ \lambda(t) \left\| \nabla_{\mathbf{x}_{t,\bar{\mathbf{m}}}} \log p_0(\mathbf{x}_{t,\bar{\mathbf{m}}} \mid \mathbf{x}_{\bar{\mathbf{m}}}) - P_{\bar{\mathbf{m}}} s_\theta(\mathbf{x}_{t,\bar{\mathbf{m}}}, \mathbf{x}_{\mathbf{m}}) \right\|_2^2 \right] \tag{34}$$

and $q(t, \mathbf{m}, \mathbf{x}_{t,\bar{\mathbf{m}}}, \mathbf{x}_{\mathbf{m}}, \mathbf{x}_{\bar{\mathbf{m}}}) = p(t)p(\mathbf{m})p_t(\mathbf{x}_{t,\bar{\mathbf{m}}}, \mathbf{x}_{\mathbf{m}})p_0(\mathbf{x}_{\bar{\mathbf{m}}} \mid \mathbf{x}_{t,\bar{\mathbf{m}}}, \mathbf{x}_{\mathbf{m}})$.

Let $s_{\boldsymbol{\theta}}^*$ denote the global minimizer of $\mathcal{L}_{\text{DISCO}}(\boldsymbol{\theta})$. From Proof A.1, we know that

$$s_{\boldsymbol{\theta}}^*(\mathbf{x}_t) = \nabla_{\mathbf{x}_t} \log p_0(\mathbf{x}_t) \tag{35}$$

We now show that $s_{\boldsymbol{\theta}}^*$ also minimizes $L_{\text{mask}}^{\text{DISCO}}(\boldsymbol{\theta})$, for any $\gamma \geq 0$. Analogously to Proof A.1, we have that

$$L_{\text{mask}}^{\text{DISCO}}(\boldsymbol{\theta}) = \mathbb{E}_{q(t,\mathbf{m},\mathbf{x}_{t,\bar{\mathbf{m}}},\mathbf{x}_{\mathbf{m}},\mathbf{x}_{\bar{\mathbf{m}}})} \left[ \lambda(t) \left\| \nabla_{\mathbf{x}_{t,\bar{\mathbf{m}}}} \log p_0(\mathbf{x}_{t,\bar{\mathbf{m}}} \mid \mathbf{x}_{\bar{\mathbf{m}}}) - P_{\bar{\mathbf{m}}} s_\theta(\mathbf{x}_{t,\bar{\mathbf{m}}}, \mathbf{x}_{\mathbf{m}}) \right\|_2^2 \right]$$

$$= \mathbb{E}_{p(t)p(\mathbf{m})p_t(\mathbf{x}_{t,\bar{\mathbf{m}}},\mathbf{x}_{\mathbf{m}})} \left[ \lambda(t) \left\| \mathbb{E}_{p_0(\mathbf{x}_{\bar{\mathbf{m}}} \mid \mathbf{x}_{t,\bar{\mathbf{m}}},\mathbf{x}_{\mathbf{m}})} \left[ \nabla_{\mathbf{x}_{t,\bar{\mathbf{m}}}} \log p_0(\mathbf{x}_{t,\bar{\mathbf{m}}} \mid \mathbf{x}_{\bar{\mathbf{m}}}) \right] - P_{\bar{\mathbf{m}}} s_\theta(\mathbf{x}_{t,\bar{\mathbf{m}}}, \mathbf{x}_{\mathbf{m}}) \right\|_2^2 \right] + \text{const.}$$

Note that the perturbation only acts on the coordinates where $\bar{\mathbf{m}}$ is 1 (i.e., the masked coordinates), and hence,

$$p_0(\mathbf{x}_{t,\bar{\mathbf{m}}}, \mathbf{x}_{\bar{\mathbf{m}}}, \mathbf{x}_{\mathbf{m}}) = p_0(\mathbf{x}_{\bar{\mathbf{m}}}, \mathbf{x}_{\mathbf{m}}) \, p_0(\mathbf{x}_{t,\bar{\mathbf{m}}} \mid \mathbf{x}_{\bar{\mathbf{m}}}). \tag{36}$$

Using this fact, we have

$$\mathbb{E}_{p_0(\mathbf{x}_{\bar{\mathbf{m}}} \mid \mathbf{x}_{t,\bar{\mathbf{m}}},\mathbf{x}_{\mathbf{m}})} \left[ \nabla_{\mathbf{x}_{t,\bar{\mathbf{m}}}} \log p_0(\mathbf{x}_{t,\bar{\mathbf{m}}} \mid \mathbf{x}_{\bar{\mathbf{m}}}) \right]$$

$$= \int p_0(\mathbf{x}_{\bar{\mathbf{m}}} \mid \mathbf{x}_{t,\bar{\mathbf{m}}}, \mathbf{x}_{\mathbf{m}}) \, \nabla_{\mathbf{x}_{t,\bar{\mathbf{m}}}} \log p_0(\mathbf{x}_{t,\bar{\mathbf{m}}} \mid \mathbf{x}_{\bar{\mathbf{m}}}) \, \mathrm{d}\mathbf{x}_{\bar{\mathbf{m}}}$$

$$= \int \frac{p_0(\mathbf{x}_{t,\bar{\mathbf{m}}}, \mathbf{x}_{\bar{\mathbf{m}}}, \mathbf{x}_{\mathbf{m}})}{p_0(\mathbf{x}_{t,\bar{\mathbf{m}}}, \mathbf{x}_{\mathbf{m}})} \cdot \frac{\nabla_{\mathbf{x}_{t,\bar{\mathbf{m}}}} p_0(\mathbf{x}_{t,\bar{\mathbf{m}}} \mid \mathbf{x}_{\bar{\mathbf{m}}})}{p_0(\mathbf{x}_{t,\bar{\mathbf{m}}} \mid \mathbf{x}_{\bar{\mathbf{m}}})} \, \mathrm{d}\mathbf{x}_{\bar{\mathbf{m}}}$$

$$= \frac{1}{p_0(\mathbf{x}_{t,\bar{\mathbf{m}}}, \mathbf{x}_{\mathbf{m}})} \int p_0(\mathbf{x}_{\bar{\mathbf{m}}}, \mathbf{x}_{\mathbf{m}}) \, \nabla_{\mathbf{x}_{t,\bar{\mathbf{m}}}} p_0(\mathbf{x}_{t,\bar{\mathbf{m}}} \mid \mathbf{x}_{\bar{\mathbf{m}}}) \, \mathrm{d}\mathbf{x}_{\bar{\mathbf{m}}}$$

$$= \frac{1}{p_0(\mathbf{x}_{t,\bar{\mathbf{m}}}, \mathbf{x}_{\mathbf{m}})} \nabla_{\mathbf{x}_{t,\bar{\mathbf{m}}}} \int p_0(\mathbf{x}_{\bar{\mathbf{m}}}, \mathbf{x}_{\mathbf{m}}) \, p_0(\mathbf{x}_{t,\bar{\mathbf{m}}} \mid \mathbf{x}_{\bar{\mathbf{m}}}) \, \mathrm{d}\mathbf{x}_{\bar{\mathbf{m}}}$$

$$= \frac{1}{p_0(\mathbf{x}_{t,\bar{\mathbf{m}}}, \mathbf{x}_{\mathbf{m}})} \nabla_{\mathbf{x}_{t,\bar{\mathbf{m}}}} p_0(\mathbf{x}_{t,\bar{\mathbf{m}}}, \mathbf{x}_{\mathbf{m}}) = \nabla_{\mathbf{x}_{t,\bar{\mathbf{m}}}} \log p_0(\mathbf{x}_{t,\bar{\mathbf{m}}} \mid \mathbf{x}_{\mathbf{m}})$$

which shows that the projection $P_{\bar{\mathbf{m}}} s_\theta(\mathbf{x}_{t,\bar{\mathbf{m}}}, \mathbf{x}_{\mathbf{m}})$ regresses the conditional score $\nabla_{\mathbf{x}_{t,\bar{\mathbf{m}}}} \log p_0(\mathbf{x}_{t,\bar{\mathbf{m}}} \mid \mathbf{x}_{\mathbf{m}})$.

Since $s_{\boldsymbol{\theta}}^*$ obeys

$$P_{\bar{\mathbf{m}}} s_{\boldsymbol{\theta}}^*(\mathbf{x}_{t,\bar{\mathbf{m}}}, \mathbf{x}_{\mathbf{m}}) = \nabla_{\mathbf{x}_{t,\bar{\mathbf{m}}}} \log p_0(\mathbf{x}_{t,\bar{\mathbf{m}}}, \mathbf{x}_{\mathbf{m}}) = \nabla_{\mathbf{x}_{t,\bar{\mathbf{m}}}} \log p_0(\mathbf{x}_{t,\bar{\mathbf{m}}} \mid \mathbf{x}_{\mathbf{m}}) \tag{37}$$

for any mask $\mathbf{m}$, this implies that $s_{\boldsymbol{\theta}}^*$ also minimizes $L_{\text{mask}}^{\text{DISCO}}(\boldsymbol{\theta})$. $\qquad\square$

## A.4 TIME-INDEPENDENT DIFFUSION MODELS

We show that minimizing $\mathcal{L}_{\text{DM}}$ with a *time-independent* score model $s_{\boldsymbol{\theta}}(\mathbf{x}_t)$, i.e.,

$$\mathcal{L}_{\text{DM}}(\boldsymbol{\theta}) = \mathbb{E}_{t,\mathbf{x}_0,\mathbf{x}_t} \left[ \lambda(t) \left\| \nabla_{\mathbf{x}_t} \log p_t(\mathbf{x}_t \mid \mathbf{x}_0) - s_{\boldsymbol{\theta}}(\mathbf{x}_t) \right\|_2^2 \right], \tag{38}$$

leads to a minimizer $s_{\boldsymbol{\theta}}^*(\mathbf{x}_t) = \mathbb{E}_{p(t \mid \mathbf{x}_t)} \left[ \nabla_{\mathbf{x}_t} \log p_t(\mathbf{x}_t) \right]$ when $\lambda(t) = 1$ and $\alpha(t) = 1$ for all $t$.

*Proof.* As the proof looks almost identical to Proof A.2, we will only briefly sketch it and refer the reader to Sun et al. (2025) for more details. With $R(\mathbf{x}_t, t) := \mathbb{E}_{p_t(\mathbf{x} \mid \mathbf{x}_t)} \left[ \nabla_{\mathbf{x}_t} \log p_t(\mathbf{x}_t \mid \mathbf{x}) \right]$, we again have that

$$s_{\boldsymbol{\theta}}^*(\mathbf{x}_t) = \mathbb{E}_{p(t \mid \mathbf{x}_t)} \left[ R(\mathbf{x}_t, t) \right] = \mathbb{E}_{p(t \mid \mathbf{x}_t)p(\mathbf{x} \mid \mathbf{x}_t)} \left[ \frac{\mathbf{x} - \mathbf{x}_t}{\sigma(t)^2} \right] \tag{39}$$

Again via Tweedie's formula, we obtain

$$s_{\boldsymbol{\theta}}^*(\mathbf{x}_t) = \mathbb{E}_{p(t \mid \mathbf{x}_t)} \left[ \frac{\mathbf{x}_t + \sigma(t)^2 \nabla_{\mathbf{x}_t} \log p_t(\mathbf{x}_t) - \mathbf{x}_t}{\sigma(t)^2} \right] = \mathbb{E}_{p(t \mid \mathbf{x}_t)} \left[ \nabla_{\mathbf{x}_t} \log p_t(\mathbf{x}_t) \right] \tag{40}$$

which concludes the proof. $\qquad\square$

# B  POSTERIOR SAMPLING

When optimizing $\mathcal{L}_{\mathrm{DISCO}}$, we need to draw samples from the $t = 0$ posterior

$$p_0(\mathbf{x} \mid \mathbf{x}_t) = \frac{q_0(\mathbf{x}_t \mid \mathbf{x}) \, p_d(\mathbf{x})}{p_0(\mathbf{x}_t)}.$$

When we set $p_d(\mathbf{x}) = p_{\mathrm{emp}}(\mathbf{x})$, we can draw *exact* samples from $p_0(\mathbf{x} \mid \mathbf{x}_t)$: Given $\mathbf{x}_t \in \mathbb{R}^D$, we compute $q(\mathbf{x}_t \mid \mathbf{x}^{(i)})$ for each $\mathbf{x}^{(i)} \in \mathcal{D}$, and sample $\mathbf{x}$ from the normalized mass function over elements in $\mathcal{D}$. Intuitively, since the perturbation kernel $q(\mathbf{x}_t \mid \mathbf{x}^{(i)})$ is an isotropic Gaussian, it will assign more probability mass to points $\mathbf{x}^{(i)}$ that are close to $\mathbf{x}^{(i)}$. This is distinct but reminiscent of the popular (minibatch) optimal transport techniques in the flow matching literature (Tong et al., 2023).

Sampling from the posterior in this way needs $O(ND)$ operations, where $N = |\mathcal{D}|$. In our low-dimensional experiments ($N = 100,000$ and $D = 2$) we draw exact posterior samples and do not observe any significant slowdown during model training. In our high-dimensional experiments (CIFAR-10, FFHQ-64), we draw approximate posterior samples by using minibatches of size 224.

Future work may explore utilizing techniques like *Locality Sensitive Hashing* (Gionis et al., 1999) or $k$-d Trees to efficiently get the $k$ nearest neighbors of $\mathbf{x}_t$, and then compute the mass function over just these neighbors. If $\sigma(0)$ is sufficiently small, this will be a good approximation to the true posterior mass function over all elements in $\mathcal{D}$.

# C  EXPERIMENTAL DETAILS

## C.1  LOW-DIMENSIONAL SETTING

**DISCO Samples.**  As discussed in Section 2.1, we might use *unconstrained* or *energy-based* score models for training DISCO, which influences our options for sampling. For unconstrained scores, we might use Unadjusted Langevin Algorithm (ULA) to draw samples from our model. However, since it is well known that Langevin algorithms suffer from slow mixing times if the target distribution is multimodal, we employ *tempering* strategies (Neal, 1996) by considering a sequence of distributions $\{p_{\beta_i}\}_{i=0}^n$ with

$$p_{\beta_i}(\mathbf{x}) \propto p_{\boldsymbol{\theta}}(\mathbf{x})^{\beta_i} \tag{41}$$

where $0 = \beta_0 < \cdots < \beta_n = 1$ is a schedule of *inverse* temperature parameters. As $\beta \to 0$, $p_\beta$ approaches a uniform distribution, and as $\beta \to 1$, we recover the original model $p_{\boldsymbol{\theta}}$. Tempering simply scales the score, i.e., $\nabla_{\mathbf{x}} \log p_\beta(\mathbf{x}) = \beta \nabla_{\mathbf{x}} \log p_{\boldsymbol{\theta}}(\mathbf{x})$. In the same way, we can also temper any *conditional* distribution of $p_{\boldsymbol{\theta}}$ given by (5).

For sampling from the DISCO model, we use *BlackJAX* (Cabezas et al., 2024) and apply tempered sequential Monte Carlo (SMC) (Naesseth et al., 2019; Doucet et al., 2001; Chopin et al., 2020; Del Moral et al., 2006) with an adaptive schedule for the inverse temperatures $\beta_i$, using an ESS target of 0.75. We perform systematic resampling after 2 Hamiltonian Monte Carlo (HMC) steps, using 3 leapfrog integration steps each. Since $\beta_0 = 0$, we initialize the SMC sampler with the uniform distribution between $-2.5$ and $2.5$.

**Wasserstein-1 Distance.**  To evaluate model fit, we sample from the model $10,000$ times, sample from $p'_d$ $10,000$ times, and compute (discrete) $W_1$ distance between them. We repeat this 30 times and report mean $\pm$ two times the standard deviation. To evaluate inference quality, we draw $10^6$ samples from the (joint) model. We then sample 30 times from $p'_d(x_1)$ and for each $x_1$, we use rejection sampling (with $\epsilon = 0.02$) to slice out the conditional samples from the joint samples. We then sample from the conditional model (e.g. using a heuristic) and again compute $W_1$ between these sets. Finally, we report the mean $W_1$ distance $\pm$ two times the standard deviation.

**Diffusion Model Samples.**  To sample from the (joint) diffusion model, we use the ancestral sampling scheme with 100 steps. This is roughly equivalent in terms of computational cost than sampling from the DISCO model.

**Weighting Function.** In all low-dimensional experiments, we let $\log(t) \sim \mathcal{U}(\log(\sigma(0)), \log(\sigma(T)))$ and use $\lambda(t) = 1$ for all $t$.

**Network Architecture.** The network architectures of the diffusion models and DISCO models are identical, except that the diffusion models receive the noise level $\sigma(t)$ as input, while the DISCO models do not. In the former case, we use a simple positional embedding for $\sigma(t)$, which we concatenate to the input. Moreover, following Tancik et al. (2020), we also use the same positional embedding for each coordinate in the input $\mathbf{x}$. The remainder of the MLP consists of 4 blocks (with residual connections), where each block contains 2 affine layers followed by leaky ReLU activations, and normalization layers at the start of each block, after the first affine layer (`InstanceNorm++` introduced in Song & Ermon (2019)). All affine layers in these blocks are maps $\mathbb{R}^K \to \mathbb{R}^K$, where we choose $K = 128$ when $D = 2$. The final block is followed by the same normalization and activation layers, and a final affine layer mapping from $\mathbb{R}^K$ to $\mathbb{R}^D$. When parameterizing the score directly, we use the output of the final hidden layer $\mathbf{z}$ as our score approximation. When building an energy-based model, we follow Du et al. (2023) and compute the energy $E_{\boldsymbol{\theta}}$ as $-\|\mathbf{z}\|_2^2$.

### C.2 High-Dimensional Setting

**Training.** To train both *DISCO* and the *EDM Masked* model, we follow the EDM repository[4] and use the same training hyperparameters of `edm-cifar10-32x32-cond-vp` and `edm-ffhq-64x64-uncond-vp`, respectively—except for the following changes: We set the Adam hyperparameter $\beta_2 = 0.95$ and use a batch size of $B = 448$ (CIFAR) and $B = 224$ (FFHQ). Each batch is constructed by sampling $B/2$ clean images, and using the same images for both the unmasked and the masked loss (sampling fresh masks every time).

**Unconditional Sampling.** We follow Karras et al. (2022) and their repository and use the same sampling settings for both CIFAR and FFHQ, except that we set $\rho = 2.5$ (for time discretization) in the DISCO CIFAR samples.

**Inpainting.** We use the same sampling setup as in unconditional sampling, except for the following changes: Sampling from the CIFAR-10 DISCO model, we use $\rho = 1.5$ for *Narrow* and *Wide* masks, $\rho = 1.8$ for *Altern. Lines* and *Super-Resolve 2x*, and $\rho = 7$ for *Half* and *Expand*. In the FFHQ DISCO model, we use $\rho = 10^3$ for *Super-Resolve 2x*, $\rho = 10^6$ for *Narrow*, *Half*, *Altern. Lines*, and $\rho = 7$ for *Wide* and *Expand*.

The *RePaint* heuristic uses a jump size of 1 and 4 resampling steps, while adjusting the outer loop iteration count such that the total number of function evaluations is the same.

## D LLM Usage

We have used LLMs to aid in the writing of this paper, including summarizing paragraphs, formatting tables, and refactoring notation. In no way have LLMs played a significant role in research ideation and writing that the LLM could be regarded as a contributor.

## E Additional High-Dimensional Qualitative Results

---

[4] https://github.com/NVlabs/edm

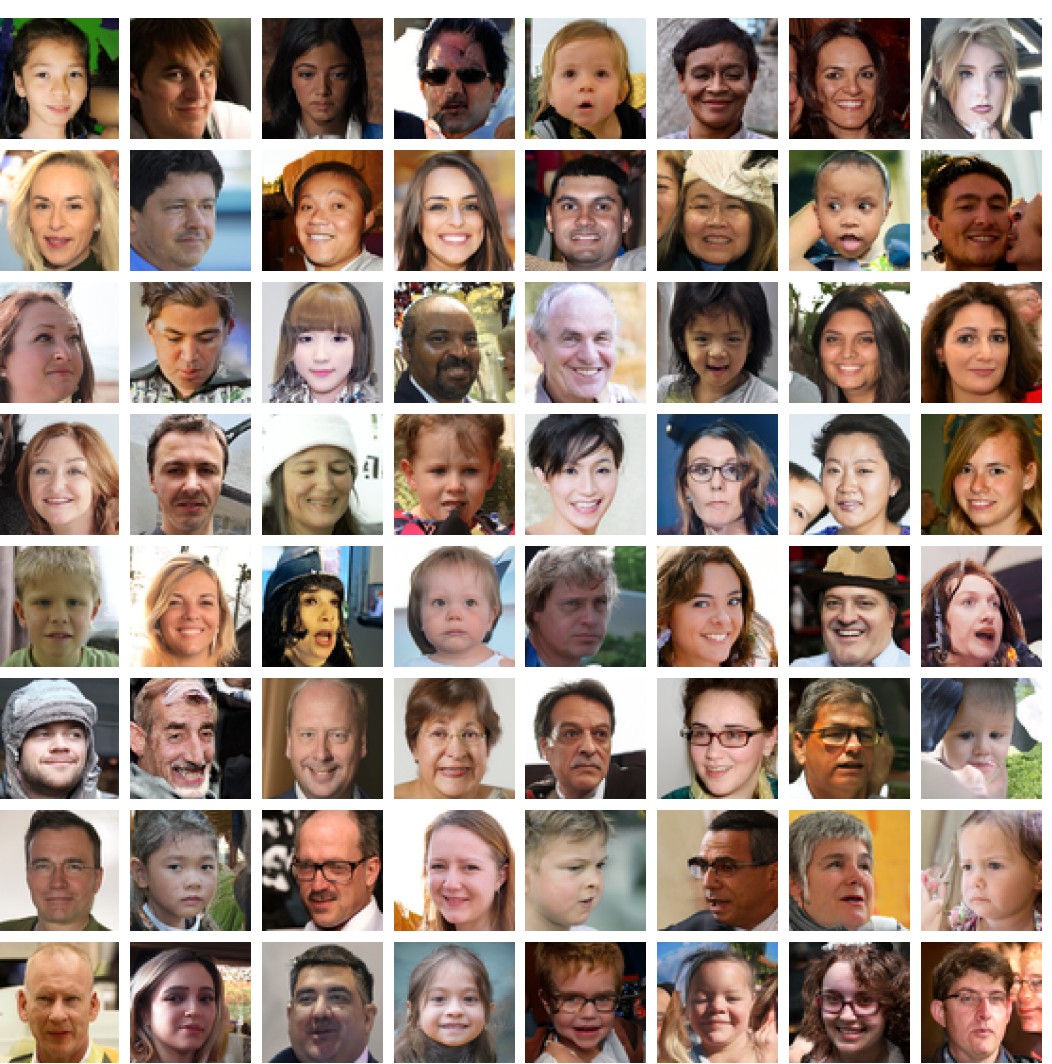

Figure 4: Unconditional samples from the DISCO model trained on FFHQ-64.

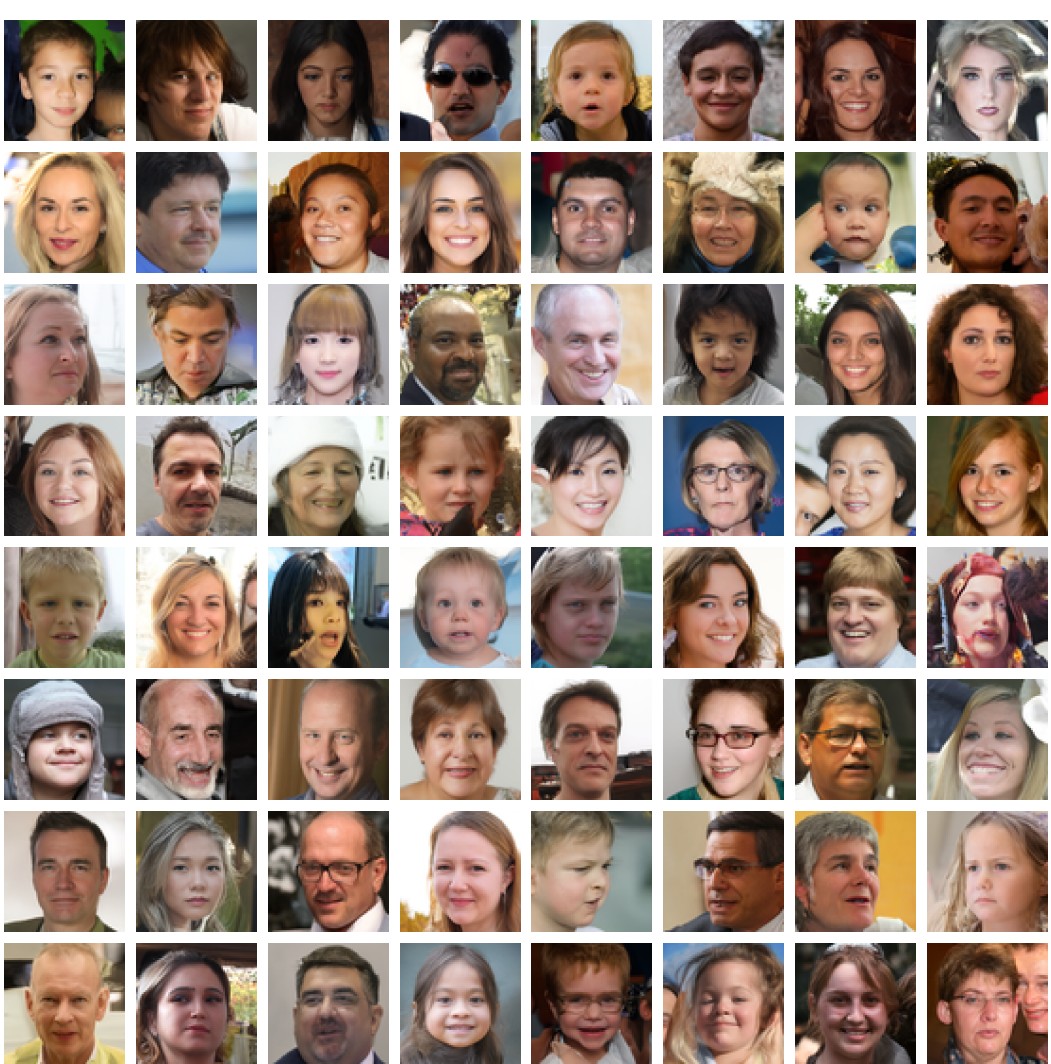

Figure 5: Unconditional samples from the diffusion model trained on FFHQ-64.

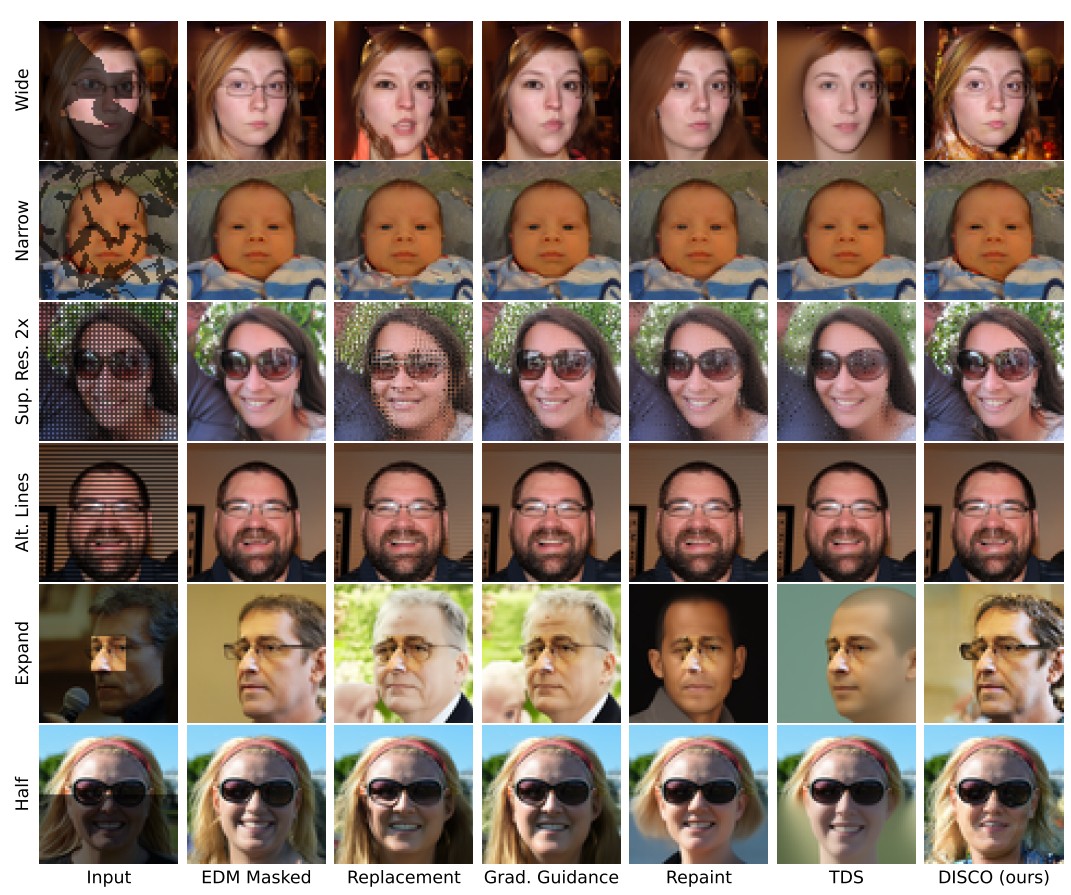

Figure 6: Qualitative inpainting results on FFHQ-64.

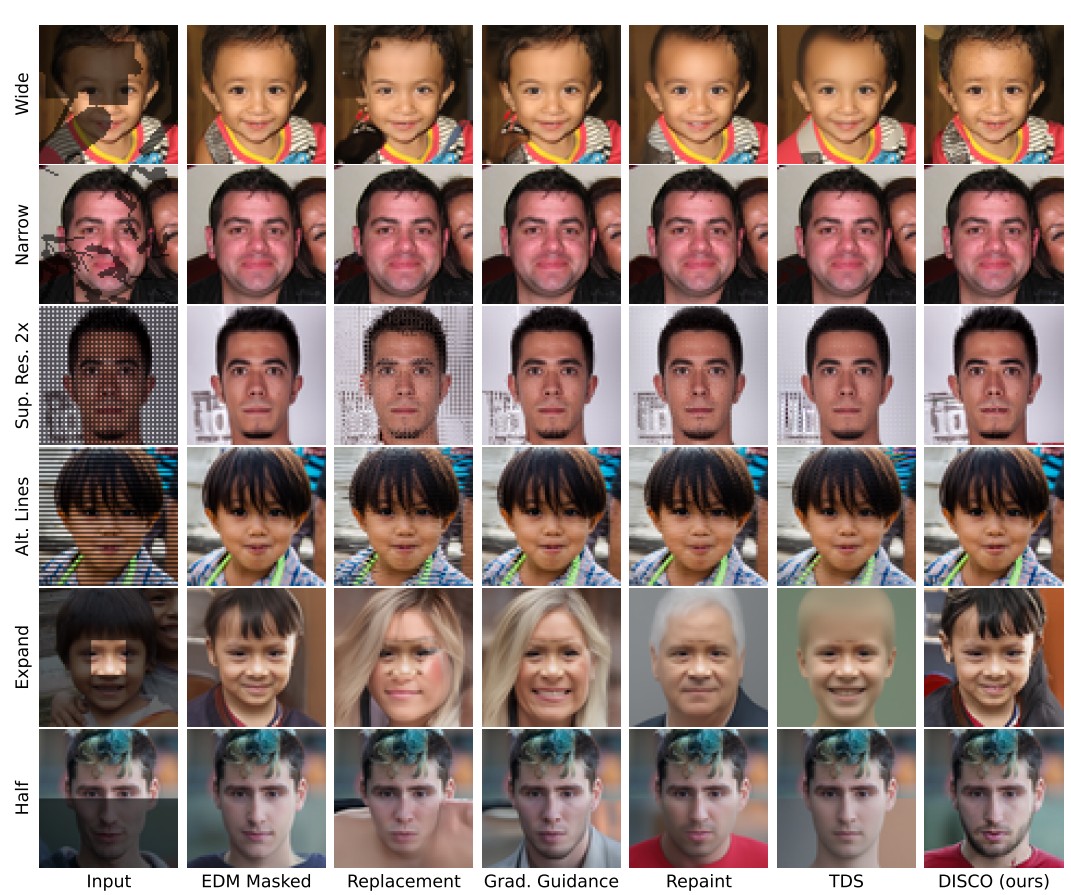

Figure 7: Qualitative inpainting results on FFHQ-64.

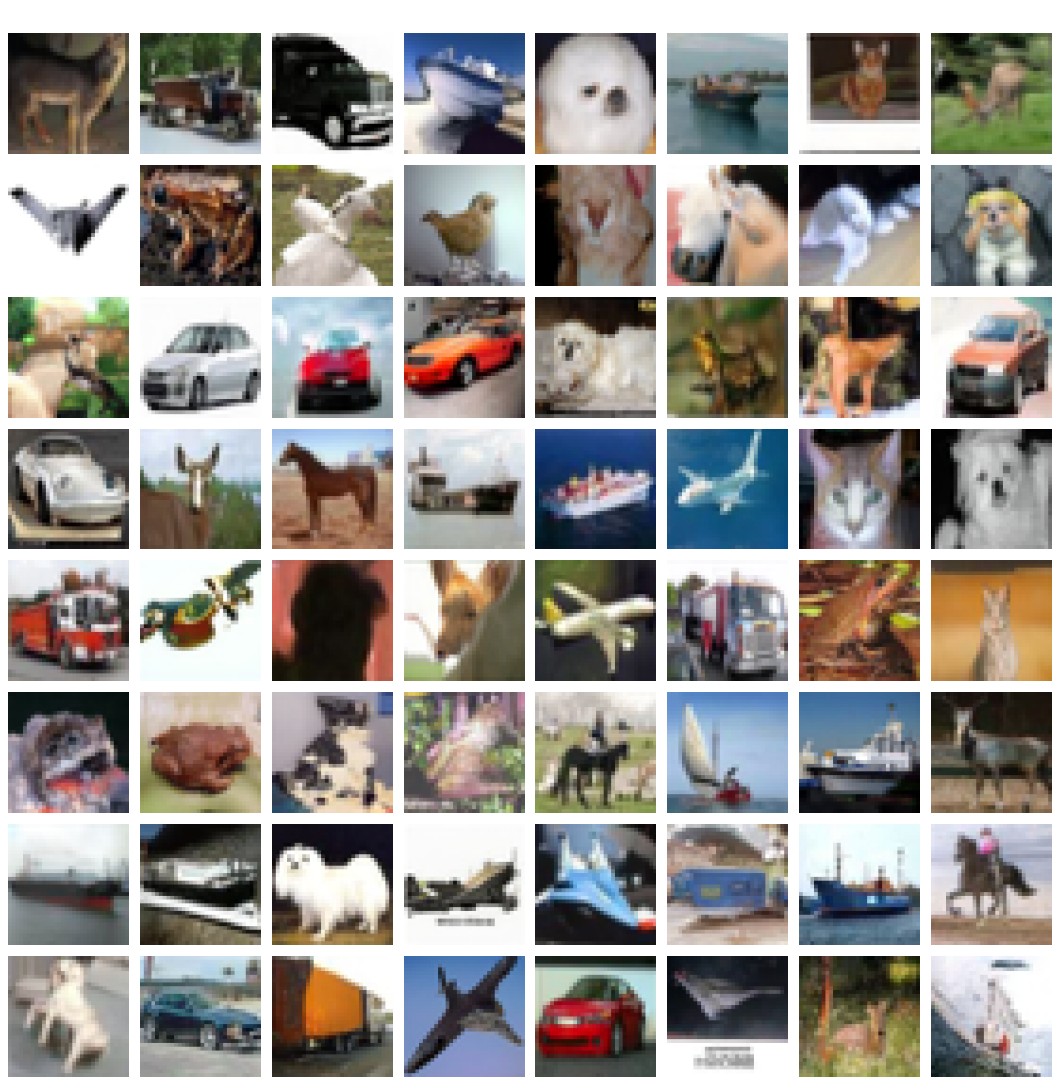

Figure 8: Unconditional samples from the DISCO model trained on CIFAR-10.

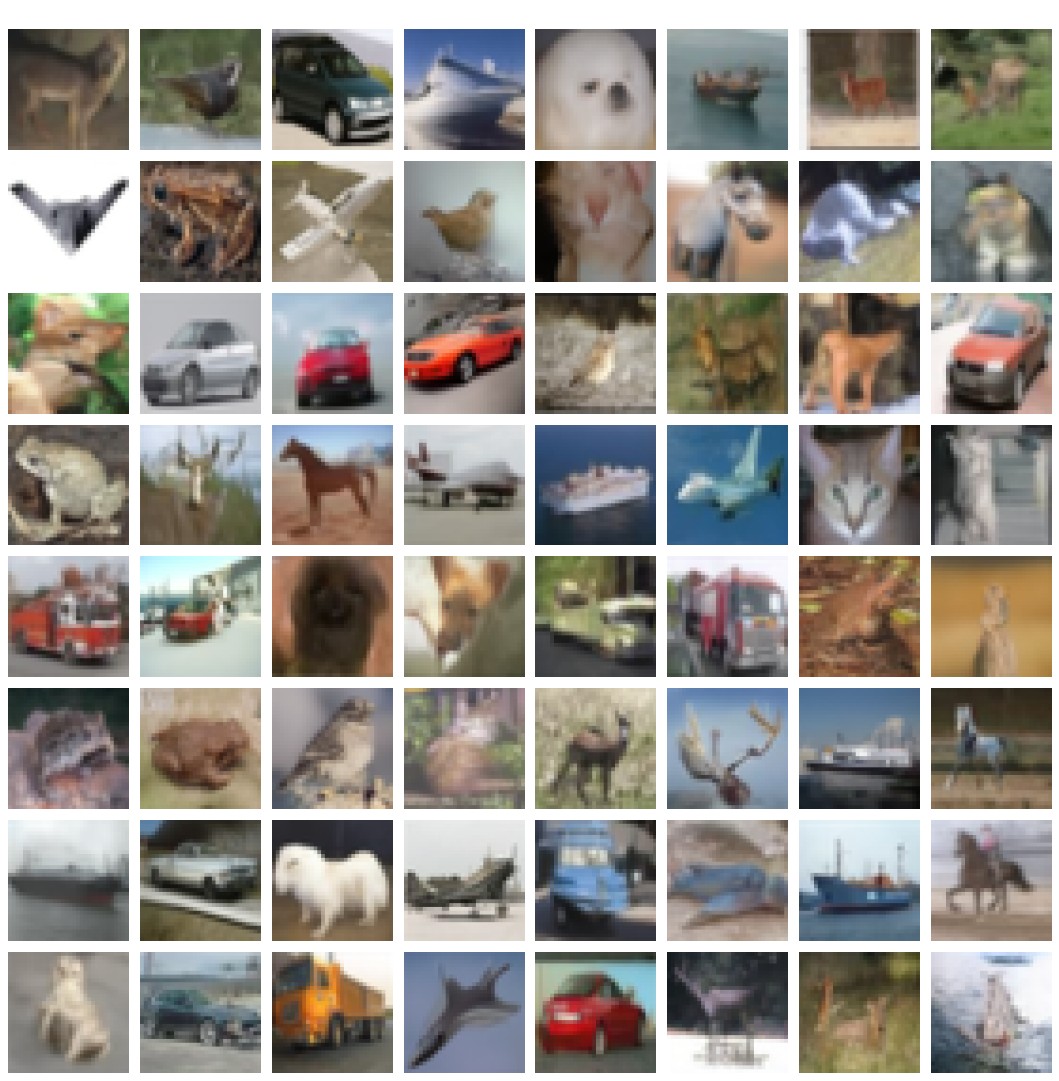

Figure 9: Unconditional samples from the diffusion model trained on CIFAR-10.

