# OpenReview forum: "Effective Diffusion-free Score Matching for Exact Conditional Sampling"
_ICLR.cc/2026/Conference — Submitted to ICLR 2026_

### Official Review · Reviewer_VWDi · 2025-10-20

**Soundness:** 2
**Presentation:** 3
**Contribution:** 2
**Rating:** 2
**Confidence:** 4

**Summary:**

This paper proposes a method for exact conditional sampling from diffusion models. The key contribution is the "DISCO loss", whose key properties are as follows:
- The global minimizer is the score of the data distribution perturbed by a small amount of Gaussian noise, and
- the score model is evaluated at all noise levels, so that the learned score is meaningful in practice away from the support of the data distribution.

The authors theoretically justify their method by showing that their DISCO loss' parameter gradients are identical to those of another loss whose minimizer is the score of the data distribution on the support of the training distribution. They show that their method significantly outperforms reasonable baselines on conditional inference from 2D toy distributions, and achieves acceptable, but generally not SOTA, performance on unconditional sampling and inpainting for image datasets.

**Strengths:**

- The paper is generally well-written, and I was able to follow the main ideas without much trouble.
- The theoretical results in this paper are correct to my knowledge.
- The proposed method performs well at conditional sampling on 2D toy datasets.

**Weaknesses:**

My primary critique of this work is that the experiments are somewhat misaligned with the paper's objectives. In the introduction, the authors claim that "diffusion models fundamentally struggle as probabilistic reasoners" and pose their central question as follows: "Can we develop score-based models that serve as sound probabilistic reasoners, providing access to exact marginals and conditionals?" To motivate this question, I would have liked to see experiments in which the authors use their method to solve a concrete probabilistic reasoning task that would otherwise be intractable. However, the authors only conclusively demonstrate their method's effectiveness for conditional sampling on 2D toy datasets; their method generally underperforms the "EDM Masked" baseline in their image experiments. I'm left with the impression that the method is currently a solution in search of a problem.

It also seems to me like the key observation is that one obtains the same solution to the score-matching problem if one takes the expectation of the $L_2$ loss over a proposal distribution whose support includes the support of the true target distribution. It is unclear to me that the proposal distribution $q(t,x,x_t)$ in the DISCO loss is better than other reasonable baselines. (e.g. what about the uniform distribution over a ball containing the support of the data distribution?) Is $q(t,x,x_t)$ a provably optimal proposal distribution? Does it outperform reasonable alternatives in practice?

**Questions:**

- Must one use $q(t,x,x_t)$ as the proposal distribution in the DISCO loss? Is it provably optimal in any reasonable sense? Does it outperform reasonable alternatives (e.g. the uniform distribution over a ball containing the support of the data distribution) in practice?
- Do the authors have any concrete probabilistic reasoning tasks in mind that would be intractable without a time-independent diffusion model trained with the DISCO loss?

---

> ### Author Response · Authors · 2025-12-03
>
> We thank the reviewer for their comments and are happy that our paper is regarded as well-written and sound, and that our approach is performing well empirically.
>
> ## Weaknesses & Questions
>
> + **Experiments are misaligned with the paper's objective**: The motivation behind DISCO is to provide a joint probabilistic model where conditional inference (via sampling) is tractable and asymptotically exact. All of our experiments are conditional sampling tasks, which are exactly aligned with this motivation. Moreover, all conditional sampling tasks we perform *are intractable* in a vanilla diffusion model, and must be approximated via heuristics.
> + **High-dimensional experiments**: Note that the baseline "EDM Masked" was trained on all conditionals, and does *not* admit a joint probabilistic model that is consistent with the learned conditionals. Meanwhile, DISCO (1) performs almost on par,  (2) provides conditional distributions consistent with its joint, and (3) even yields better unconditional samples on FFHQ-64.
> + **The method is a solution in search of a problem**: Conditional inference in score-based models is an important and highly active field of research, and contemporary heuristics fail even in low-dimensional toy settings. We disagree strongly with this statement.
> + **Proposal distribution**: Technically, the mixture $\sum_t \lambda(t) p_t(\mathbf{x}_t)$ is the "proposal distribution", not $q(t, \mathbf{x}, \mathbf{x}_t)$. If $q(t, \mathbf{x}, \mathbf{x}_t)$ is *not* chosen as in Theorem 1, then the network will in general *not* learn the scores of $p_d'$ when minimizing $\mathcal{L}\_{\text{DISCO}}$---no matter the choice of proposal distribution. The choice of $p_t$ as Gaussian mixtures is neither provably optimal in vanilla diffusion models, nor in DISCO. While Gaussian noise is computationally convenient to simulate from, future work may extend this by considering other proposal distributions (akin to non-Gaussian diffusion models).

---

### Official Review · Reviewer_Qg4t · 2025-10-25

**Soundness:** 3
**Presentation:** 2
**Contribution:** 3
**Rating:** 4
**Confidence:** 3

**Summary:**

The paper proposes a method to learn the score of a target distribution using only samples from it. The approach minimizes a weighted mixture of Fisher divergences, extending the principle of denoising score matching (DSM). Instead of using a single noise level as in classical DSM, the method averages over multiple perturbed distributions with varying noise magnitudes. This ensures accurate score estimation both in high-probability regions (captured by small-noise perturbations) and in low-probability regions (captured by larger noise levels). The approach is validated on 2D toy distributions and image modeling tasks.

**Strengths:**

* The paper addresses an important and timely problem.
* The work demonstrates careful attention to detail, notably in the treatment of posterior sampling, a subtle and technically challenging aspect often overlooked in prior studies.

**Weaknesses:**

* The paper is poorly written and often difficult to follow.
* Although the paper mentions training energy-based models, such experiments are never actually conducted. This omission is critical, as estimating the Fisher divergence in low-probability regions lies at the heart of the score-matching blindness problem, which affects many applications (see [1,2,3]). Moreover, the proposed approach is closely related to existing methods that attempt to address this issue (see [2] and, more recently, [4]). This connection should be emphasized, and comparisons with these approaches would substantially strengthen the paper.
* The work clearly aims to address the limitations of MDSM, yet this motivation appears only briefly in the related work section and Appendix A.2. The discussion in Appendix A.2 should be brought into the main text to better justify the proposed method and clarify the role of the posterior sampling component, which is currently difficult to understand.
* As acknowledged by the reviewers in Appendix B, the empirical approximation of the data distribution in posterior sampling poses significant computational and statistical challenges.
* A major weakness lies in the choice of the weighting function $\lambda(t)$. While averaging over multiple noise levels is conceptually appealing, the variance of the objectives can differ greatly across values of $\sigma(t)$, making optimization unstable and difficult.

[1] Wenliang, L., & Kanagawa, H.. (2021). Blindness of score-based methods to isolated components and mixing proportions.

[2] Zhang, M., Key, O., Hayes, P., Barber, D., Paige, B., & Briol, F.X. (2022). Towards Healing the Blindness of Score Matching. In NeurIPS 2022 Workshop on Score-Based Methods.

[3] Shi, Z., Yu, L., Xie, T., & Zhang, C.. (2024). Diffusion-PINN Sampler.

[4] Tobias Schröder, Zĳing Ou, Jen Ning Lim, Yingzhen Li, Sebastian Josef Vollmer, & Andrew Duncan (2023). Energy Discrepancies: A Score-Independent Loss for Energy-Based Models. In Thirty-seventh Conference on Neural Information Processing Systems.

**Questions:**

* Could you reproduce Fig. 2 (and, more generally, Section 5.1) using an energy-based parametrization? Visualizing the learned log-probability would make Fig. 2 much clearer and easier to interpret.
* Would it be possible to include experiments on synthetic Gaussian mixtures (ideally with unequal mode weights) to demonstrate whether the proposed method alleviates the score-matching blindness issue? This setup would also be fully tractable.
* Out-of-distribution detection seems like a natural application of this work, have you considered evaluating it in that context?
* It would be valuable to strengthen the connection with the energy-based modeling community, which addresses closely related problems [A,B,C,D,E]. For instance, [C] also integrates diffusion-based ideas in that setting.
* I had difficulty understanding the image experiments: how do you perform sampling with the EDM Heun sampler without a time-dependent denoiser? Do you simply ignore the time dependence during sampling?

[A] Nijkamp, E., Hill, M., Han, T., Zhu, S.C., & Wu, Y. (2020). On the Anatomy of MCMC-Based Maximum Likelihood Learning of Energy-Based Models. Proceedings of the AAAI Conference on Artificial Intelligence, 34(04), 5272–5280.

[B] Gao, R., Nijkamp, E., Kingma, D., Xu, Z., Dai, A., & Wu, Y. (2020). Flow Contrastive Estimation of Energy-Based Models. In Proceedings of the IEEE/CVF Conference on Computer Vision and Pattern Recognition (CVPR).

[C] Ruiqi Gao, Yang Song, Ben Poole, Ying Nian Wu, & Diederik P Kingma (2021). Learning Energy-Based Models by Diffusion Recovery Likelihood. In International Conference on Learning Representations.

[D] Nijkamp, E., Gao, R., Sountsov, P., Vasudevan, S., Pang, B., Zhu, S.C., & Wu, Y. (2022). MCMC Should Mix: Learning Energy-Based Model with Neural Transport Latent Space MCMC. In International Conference on Learning Representations.

[E] Grenioux, L., Moulines, E., & Gabrie, M. (2023). Balanced Training of Energy-Based Models with Adaptive Flow Sampling. In ICML 2023 Workshop on Structured Probabilistic Inference & Generative Modeling.

---

> ### Author Response · Authors · 2025-12-03
>
> We are happy that our work is regarded as important and detailed. However, we believe that the reviewer has not understood the main parts of the paper:
>
> ## Weaknesses
>
> + **"Although the paper mentions training energy-based models, such experiments are never actually conducted."**: This is absolutely not true. The *first paragraph* in Section 5 (Experiments) reads "[...], we train both a regular energy-based diffusion model and an energy-based DISCO model [...]". The captions of Figure 1 and Figure 2 re-iterate this fact.
> + **Score-matching blindness**: In [2], the authors minimize Fisher divergence between the model and a 2-component mixture distribution, where only one component is of interest. To remove this component after training, one must estimate the model's normalizing constant, which is intractable in high dimensions. Our work is very different because we (1) minimize *weighted* Fisher divergence, (2) learn the score of the desired target $p_d'$, and (3) do not have to ever estimate normalizing constants. We will comment on the blindness issue in the revised paper.
> + **MDSM**: We will move some of the discussion in Appendix A.2 into the main text.
> + **Posterior Sampling**: We again note that since naive sampling is linear in the number of data samples, it is straightforward in the low/medium data regime, and can be easily approximated (e.g. via mini-batches) in the large data regime.
> + **Weighting Function**: It is not clear why the reviewer regards the weighting function as a "major weakness": It serves the same role as in traditional diffusion training, and we have not observed significantly more difficult or unstable optimization when training DISCO.
>
> ## Questions
>
> + **Figure 2**: The scores visualized in Figure 2 do stem from energy-based models, as noted in the caption.
> + **GMM Experiments**: It is not clear how the reviewer has missed Figure 1, which exactly shows the proposed experiment.
> + **Out-of-distribution (OOD) detection**: We have not conducted experiments on OOD detection, since this is orthogonal to our main motivation.
> + **Energy-based models**: We do not agree that the cited papers address "closely related problems", since none of them study conditional sampling from a learned joint distribution.
> + **EDM Sampler**: Yes, we use the Heun sampler without modifications and replace time-dependent diffusion model with our DISCO model. The revised version will mention this fact into the main text.

---

### Official Review · Reviewer_oy6K · 2025-10-25

**Soundness:** 2
**Presentation:** 2
**Contribution:** 2
**Rating:** 2
**Confidence:** 4

**Summary:**

Summary

The goal is to draw samples from *conditional* distributions of data, having sample-access to the *joint* distribution of data. The authors do so in three steps:

**Step 1**. Estimate the score vector of the *joint* distribution

**Step 2**. Deduce the score vector of the *conditional* distribution (discard the vector entries you condition on, Eq. 5)

**Step 3**. Plug that score into a Sequential Monte Carlo *sampler*

The authors' contribution is in step 1: they propose a new loss function for estimating the score, called DISCO.

**Strengths:**

### **Original Problem Statement**

The original score-matching loss measures the squared error between the true and model scores *in-distribution*, that is, averaged over samples drawn from the *data* distribution. Much of the prior literature has focused on reformulating this loss into computationally tractable forms [1, 2].

In contrast, the authors propose to evaluate the squared error between the true and model scores *out-of-distribution*, that is, averaged over samples drawn from *other* distributions. In practice, these distributions are “flattened” versions of the data density, simulated by adding Gaussian noise to data samples.

Out-of-distribution estimation of the data score is an important topic and the authors' formulation is interesting. For example, sampling from the product of two distributions can be done by adding their scores [3].  In such a case, the scores are typically trained using samples from individual distributions rather than their product, which can lead to inaccuracies when evaluated on samples from the product. The authors’ approach could address this mismatch.

[1] Hyvärinen, Estimation of Non-Normalized Statistical Models by Score Matching, JMLR 2005.

[2] Vincent, A Connection Between Score Matching and Denoising Autoencoders, Technical Report 2010.

[3] Du et al., Reduce, Reuse, Recycle: Compositional Generation with Energy-Based Diffusion Models and MCMC, ICML 2023.

**Weaknesses:**

## 1. Motivation for the DISCO Loss

What is the motivation for the DISCO loss (Eq. 6), which estimates the data score *out-of-distribution*?

The only reason I can think of is that the authors anticipate the SMC sampler will query the score function *outside* the data distribution, and therefore the authors would like the score to be accurately estimated in these areas. If this is indeed the motivation, it should be stated explicitly, since it is the starting point of the entire contribution.

## 2. Concern with the Loss Function

The main technical contribution is the DISCO loss (Eq. 13) for estimating the score of the joint distribution. However, evaluating this loss requires sampling from the distribution of clean data given noisy data, also called denoising posterior (see the “DISCO Training” paragraph).

This is a difficult problem in itself! Sampling from the denoising posterior is exactly what one-step generative models (*e.g.* consistency and flow-map models) try to do. If we could sample accurately from the denoising posterior, we would essentially have solved the data generation problem.

I understand that the authors suggest that sampling from an approximation of the denoising posterior is sufficient, but this should be further discussed, especially as it raises questions about both feasibility of the training objective.

## 3. Concern with the Evaluation Procedure

As stated in the summary, the authors use a three-step process for conditional sampling:

**1-** Estimate the score of the joint distribution.

**2-** Derive the conditional score.

**3-** Plug this conditional score into an SMC sampler

There are two main sources of error in this pipeline:
- The estimation error of the score
- The sampling error from the SMC sampler

The paper’s contribution concerns the first part: improving score estimation. Therefore, the natural evaluation would be to compare different score estimators using the **same** SMC sampler. Instead, the experiments pair different estimators with **different samplers** (e.g., DISCO + SMC vs. DSM + diffusion guidance). This confounds the interpretation: improvements could very well be due to the choice of sampler rather than from better score estimation. In fact, I suspect this is the case: previous work explicitly compares SMC (Annealed Langevin Dynamics specifically) and diffusion guidance, and showed drastic difference in results that highlight that SMC is an unbiased sampler while diffusion guidance is a biased sampler [2].

Moreover, the SMC sampler is barely described in the main text: it is mentioned only once on page 3 and once on page 8. Given that it is the main sampling algorithm the authors use to generate samples from conditional distributions, it definitely deserves a more detailed explanation (e.g. choice of schedule, of initialization).

Lastly, Figure 2, which visualizes joint scores estimated by different methods, does not seem very useful. The differences between DISCO and other score estimators are not visually striking, but more importantly, the authors care that the conditional scores are correct which is not obvious to visualize on a plot that shows the joint scores.

[2] Du et al. Reduce, Reuse, Recycle: Compositional Generation with Energy-Based Diffusion Models and MCMC. ICML 2023.

## 4. Clarity in the Writing

While the paper is overall clear, some passages are quite confusing.

**Introduction**
> “This objective, albeit reminiscent of diffusion training, only fits the (slightly perturbed) data distribution rather than a full diffusion process, while taking care that the score field is also fit outside the data manifold. This approach makes conditioning de-facto trivial: one simply fixes observed variables in the learned score field and samples only the unobserved variables—enabling asymptotically exact probabilistic inference.”


**Masked DISCO Training**
> “During conditional sampling, the model must have learned the score at points $(\mathbf{x}^u, \mathbf{x}^c)$ where $\mathbf{x}^c$ is ‘clean’, and $\mathbf{x}^u$ is ‘noisy’. While in theory, minimizing $\mathcal{L}_{\text{DISCO}}$ learns the true score also at these points, we observe that in high dimensions, the model does not learn accurate scores at these points.”

It is not clear to me that the model "must" have learned the noisy score for conditional sampling. While diffusion-based samplers use the noisy conditional scores, an SMC sampler only requires the clean conditional score.

**DISCO Training and Masked DISCO Training**
These paragraphs are hard to read.

## 5. Clarifying the Related Work

The paper refers to gradient guidance (Ho et al., 2022), the replacement heuristic (Song et al., 2020), and the Twisted Diffusion Sampler (Wu et al., 2023).

However, I could not find the term “replacement heuristic” in the cited Song et al. (2020) papers. If this terminology is introduced by the authors, it should be clearly stated, and the relevant passage in the original work cited precisely.

The authors state that the Twisted Diffusion Sampler offers “only asymptotic guarantees.” Asymptotic in what sense—in the number of particles, iterations of SMC, or samples? I would note that the authors’ method is also correct only asymptotically! Exact sampling from the conditional distributions would require perfect score estimation (infinite data) and convergence of SMC (infinite iterations).

Also, the authors claims that diffusion guidance lacks theoretical guarantees. It would be relevant to cite recent theoretical analysis of guidance [1].

[1] Chidambaram et al. What Does Guidance Do? A Fine-Grained Analysis in a Simple Setting. NeurIPS 2025.

## 6. Missing Related Work

The authors ommitted a relevant and recent line of work, which uses diffusion and flow models to do unbiased sampling of conditional distributions [1–3]. In particular, (3) clearly explain how this works: one can amortize the score (or velocity) estimation across denoising schedules. To sample from $p(x^1 \mid x^2)$, one first denoises $x^1$ and then denoises $x^2$. The first part effectively samples from the conditional.

[1] Chen et al. Diffusion Forcing: Next-Token Prediction Meets Full-Sequence Diffusion. NeurIPS 2024.

[2] Song et al. History-Guided Video Diffusion. ICML 2025.

[3] Wewer et al. SRM: Spatial Reasoning with Denoising Models. ICML 2025.

**Questions:**

Can the authors address the six concerns I have detailed in the Weaknesses section?

My foremost concern is that successful sampling from the conditional distribution is attributable to the choice of sampler (SMC vs. diffusion guidance) rather than the score estimation (DISCO vs. DSM at multiple noise levels). Because the authors' main contribution is the score estimation, this deserves special attention. Can the authors comment on this?

---

> ### Author Response · Authors · 2025-12-03
>
> We thank the reviewer for their insightful comments and are happy that our approach is considered to address an important problem.
>
> However, we feel like the reviewer has misunderstood central parts of the paper:
>
> ## Weaknesses
>
> + **Motivation**: As laid out in the main text, the central motivation for DISCO is to learn a joint distribution that admits asymptotically exact and consistent access to any conditional distribution via MCMC. This can be done by learning a single score function, but naive Denoising Score Matching fails to learn the score away from the data manifold, which is detrimental for MCMC algorithms that do not start on the manifold (see discussion in Section 2.2). This is why we introduce our novel DISCO loss, which can accurately learn scores far from the manifold. This point is made very explicitly in Section 2.2, Section 2.3, and Section 3 (e.g. L179: "Instead, we aim to learn a *single* score field, which allows us to sample any conditional according to (5)").
>
> + **Loss Function**: Since the empirical distribution is the uniform distribution over data points, sampling from the denoising posterior is *linear* in the number of data points, which makes sampling from the *true* denoising posterior easy in the low/medium data regime (see Appendix B for discussion). In the large data regime, sampling can be approximated e.g. via mini-batches.
>
> + **Evaluation**
> 	+ Since tempering a distribution and diffusing it are two different operations, one cannot simply use the tempered SMC sampler in conjunction with a time-indexed diffusion model. Therefore, it is not possible to use the same SMC sampler for both DISCO and a time-dependent diffusion model.
> 	+ To address the reviewer's concern about confounding, the revised version of the paper will add experiments that use SMC to (1) sample from an energy-based diffusion model with fixed $t=0$, and (2) sample from an energy-based diffusion model that is time independent (as in Figure 2).
> 	+ We note that the metric for "Inference Quality" is independent of the model fit (i.e., how accurate the scores are), since it only measures distance between the approximate conditional distribution and the true model conditional.
> 	+ The SMC sampler (as well as our choice of schedule and initialization) is described in detail in Appendix C.1, which is referred to in the main text.
> 	+ Figure 2 clearly shows that norms of the learned scores are consistently and severely underestimated by the $t=0$ diffusion model (rightmost column), and that the $t$-independent diffusion model also learns inaccurate score norms. The purpose of Figure 2 is not to show conditional distributions, which is done in Figure 1.
>
> + **Clarity in the Writing**:
> 	+ **Introduction**: It is not clear what the reviewer's critique of this section is. Simply citing a paragraph without comment is not constructive criticism.
> 	+ **Masked DISCO Training**: "Noisy" just means "a point off the manifold". The SMC sampler *does* require accurate scores in regions where $\mathbf{x}^u$ is "noisy" and $\mathbf{x}^c$ is "clean", as it is exactly initialized on such a point (see e.g. last paragraph in Section 2.2). It seems that the confusion stems from equating "noisy" with $t \gg 0$---however, we do not have a notion of time $t$ during sampling.
> 	+ **DISCO Training and Masked DISCO Training**: The revised version will refactor these paragraphs to make them easier to read.
>
> + **Clarifying the Related Work**:
> 	+ **Replacement Heuristic**: We use the same nomenclature as in [4], and will refer to the relevant passages more explicitly in the revision.
> 	+ **Asymptotic Guarantees**: The paragraph on L366 specifies what we mean with "asymptotic guarantees", namely the number of particles.
> 	+ **Guidance**: We state that *Gradient Guidance* lacks theoretical guarantees, which is true. Note that *Gradient Guidance*, as defined in Section 4, is different from "classifier guidance" and "classifier-free guidance" in diffusion models, where the conditional score $\nabla_{\mathbf{x}_t} \log p_t(\mathbf{x}_0^c \mid \mathbf{x}_t)$ is assumed to be known.
>
> + **Related Work**: [1] is a diffusion-based sequence model, which does not support generic conditional inference. [2] and [3] directly learn conditional diffusion procedures for specific conditioning patterns and do not learn a single joint density. DISCO is fundamentally different: it learns a single, time-independent joint distribution over all variables, from which *all* conditionals can be obtained (asymptotically exactly) via standard MCMC.
>
> ----------
>
> [1] Chen et al. Diffusion Forcing: Next-Token Prediction Meets Full-Sequence Diffusion. NeurIPS 2024.
>
> [2] Song et al. History-Guided Video Diffusion. ICML 2025.
>
> [3] Wewer et al. SRM: Spatial Reasoning with Denoising Models. ICML 2025.
>
> [4] Wu et al. Practical and asymptotically exact conditional sampling in diffusion models. NeurIPS 2023

---

### Official Review · Reviewer_Q8EJ · 2025-11-02

**Soundness:** 3
**Presentation:** 2
**Contribution:** 2
**Rating:** 2
**Confidence:** 3

**Summary:**

This paper proposes a novel training paradigm to perform conditional sampling via diffusion models. The authors argue that existing methods for conditional sampling based on diffusion models are largely *heuristic*, where the objective functions being used do not correspond to well-defined minimizers, etc. To circumvent this issue, they propose DISCO, a diffusion-free score-matching objective based which learns a time-independent score function which they can use to perform conditional sampling in a principled manner. While the authors do still incorporate a notion of "noising process" at various amounts of noise, the end-goal of their approach is to only learn the score very close to the data distribution i.e., when the noise is essentially zero. The objective function being used is related to the relative Fisher information between the two distributions, as well as an additional "masking" loss. The authors compare their method with heuristic approaches on a suite of examples.

**Strengths:**

I appreciate the level of rigor that the authors want to bring to the challenging task of conditional sampling, and the detailed background of different existing methods based on diffusion models. The use of a new objective function (the reweighted Fisher divergence) also appears to be new. I also think the presentation for the experiments was well put together.

**Weaknesses:**

- Lack of discussion/comparison or citation of other methods for conditional simulation: There is a swath of literature on this topic based on optimal transport, GANs, or Schrödinger bridges, that deserves a mention or citation; see below for examples and the references therein.

- Presentation style: There are too many distributions named $p$ but correspond to different things. For example, there is $p(t)$ as well as $p(\textbf{m})$, and it is a bit confusing. The naming convention for the loss $\mathcal{L}_{\rm DISCO}^{\rm mask}$ and the version when the super and subscript are swapped might be worth revisiting.

- The method claims to be principled but I am not entirely sure I understand that it is beyond the loss-minimization aspect. The use of a new object (reweighted FI) is interesting, but the fact that the objective function itself requires conditional sampling of some kind (this is my understanding between likes 240 and 243). It is entirely possible I misunderstood something, but this does seem to be a limitation.

- I appreciate the use of Inference Quality as a metric, but the comparison seems like it could be improved. For instance, would it not be better to perform MC estimation of
$$ \mathbb{E}_{p'_d(x_1)}[ W_1( p_d(x_2|x_1) , p^\theta(x_2|x_1)], $$
where the first argument is the ground-truth conditional distribution? This can be done in the case of Gaussian distributions or other simple 2D settings (see the references below for examples).

-  The use of 10^5 examples in 2D is a bit high, given that this many samples does not even exist for CIFAR10 or MNIST.

Some references:

@article{kerrigan2024dynamic,
  title={Dynamic conditional optimal transport through simulation-free flows},
  author={Kerrigan, Gavin and Migliorini, Giosue and Smyth, Padhraic},
  journal={Advances in Neural Information Processing Systems},
  volume={37},
  pages={93602--93642},
  year={2024}
}

@article{baptista2024conditional,
  title={Conditional simulation via entropic optimal transport: Toward non-parametric estimation of conditional Brenier maps},
  author={Baptista, Ricardo and Pooladian, Aram-Alexandre and Brennan, Michael and Marzouk, Youssef and Niles-Weed, Jonathan},
  journal={arXiv preprint arXiv:2411.07154},
  year={2024}
}

@inproceedings{shi2022conditional,
  title={Conditional simulation using diffusion Schr{\"o}dinger bridges},
  author={Shi, Yuyang and De Bortoli, Valentin and Deligiannidis, George and Doucet, Arnaud},
  booktitle={Uncertainty in Artificial Intelligence},
  pages={1792--1802},
  year={2022},
  organization={PMLR}
}

**Questions:**

Why is having a family of proposal distributions a good idea as opposed to one fixed choice? This is a central question that I was not able to understand.

---

> ### Author Response · Authors · 2025-12-03
>
> We appreciate the reviewer's comments and are happy that our work is considered to be novel and well presented.
>
> However, we believe that the reviewer has misunderstood the fundamental goal of our work, which is to provide a joint distribution which admits asymptotically exact access to all possible conditional distributions. Importantly, our work is *not* about directly learning conditional distributions where no consistent joint distribution may exist.
>
> ## Weaknesses
>
> + **Lack of discussion**: The cited papers are fundamentally different from our work, since all of them *directly learn (possibly time-indexed) conditional distributions only*, without access to the joint distribution. In contrast, DISCO learns a single joint distribution that admits access to all true conditional distributions via MCMC, allowing consistent probabilistic inference. Other generative models like GANs are also not related to our work.
> + **Presentation style**: While we will refactor some notation in the revised version of the paper, it is very standard in probability theory to overload symbols like $p$ in expressions like $p(t)$ and $p(\mathbf{m})$ for conciseness. We do not believe that overloading $p$ makes the paper harder to read.
> + **Posterior Sampling**: The fact that the objective requires sampling from the $t=0$ posterior is discussed in the main text (L233), and more exhaustively in Appendix B. Since naive sampling is linear in the number of data samples, it is straightforward in the low/medium data regime, and can be easily approximated (e.g. via mini-batches) in the large data regime.
> + **Inference Quality Metric**: The use of our "Inference Quality" metric is chosen over your proposed version because it does *not* conflate model fit with inference quality. No matter how bad the model fit, our metric simply measures how close the approximate conditional distribution is to the *true model conditional*. The data distribution $p_d$ does not enter on purpose.
> + **10^5 examples in 2D**: This was an arbitrary choice and none of our methods require this large amount of samples. The revised version of the paper will include 2D experiments with less number of data points. However, we disagree with listing this as a "weakness".
>
> ## Questions
>
> Technically, one can consider our proposal distribution as a single *mixture* distribution over different noise levels. While future work may consider other distributions, this choice yields a training setup which is reminiscent of regular diffusion models and a forward process which can be simulated in a single step.

---

### Meta-Review · Area_Chair_yXnw · 2026-01-05

**Summary:**

DISCO proposes a diffusion-free, time-independent score-matching objective intended to enable asymptotically exact conditional sampling from a learned joint distribution. Multiple reviewers consider the core derivations plausible/correct and the problem well-motivated (e.g., Reviewers Q8EJ, Qg4t, VWDi), and the 2D conditional-sampling results clearly illustrate failure modes of common diffusion heuristics (e.g., Fig. 1 / Table 1; Reviewer VWDi). However, the paper’s strongest empirical support remains in low-dimensional toys, while the high-dimensional image results are mixed: DISCO is not clearly better than the strongest baselines overall, and it is notably worse on CIFAR-10 unconditional FID than EDM/EDM-Masked (Table 3; Reviewer VWDi’s “acceptable but not SOTA” summary). A central concern is experimental confounding: DISCO is paired with tempered SMC while baselines rely on different conditional samplers (often biased), so it is not yet clear how much of the gain comes from improved score estimation vs. the sampling procedure, and the rebuttal mainly promises additional controlled SMC baselines rather than presenting them (Reviewer oy6K). The feasibility and validity of the required denoising-posterior sampling $p_0(x \mid x_t)$ in high dimensions is still not convincingly demonstrated. While exact sampling is feasible for small datasets, the large-scale approximations (e.g., minibatch- or kNN-based) are not analyzed, leaving their impact on bias, variance, and scalability unclear.


Overall, even if some concerns are partially addressed in the rebuttal, the current evidence and clarity do not yet justify the paper’s broad “probabilistic reasoner / exact conditional” positioning, so I recommend **reject**.

**Reviewer Concerns:**

- Reviewer Q8EJ's main concern is that the DISCO objective requires conditional sampling, namely sampling from the denoising posterior $p_{0}(x | x_t)$, which can be a nontrivial inference problem and may limit practical tractability.

- Reviewer oy6K mainly raises three main concerns: (i) the motivation for the DISCO loss is not stated clearly; (ii) optimizing the loss requires sampling from the denoising posterior $p_{0}(x | x_t)$, which may itself be a difficult inference problem; and (iii) the evaluation confounds score-estimation error with sampling error, since the experiments pair different estimators with different samplers rather than comparing score estimators under a common sampling procedure.

- Reviewer Qg4t notes that the paper is often difficult to follow (presentation/clarity). They also highlight that DISCO training relies on posterior sampling from the empirical approximation to $p_{0}(x | x_t)$, which can introduce computational and statistical challenges in high dimensions. Finally, they question the choice of the weighting function, arguing that different noise levels may yield objectives with very different variances, potentially making optimization unstable or harder in practice.

- Reviewer VWDi finds the experiments only partially aligned with the paper's stated goal of enabling sound probabilistic reasoning (i.e., access to exact marginals/conditionals), and asks for a concrete probabilistic reasoning task that is intractable for standard diffusion models but becomes tractable under DISCO. They also question whether the proposal distribution used in the DISCO loss is justified relative to plausible alternatives (e.g., uniform over a region containing the data support), and whether it is optimal or empirically preferable beyond convenience.

**Reviewer Scores:**

Several reviewers raise concerns about clarity and motivation (e.g., Reviewer oy6K on the motivation for the proposed loss, Reviewer Qg4t on the overall presentation being hard to follow, and Reviewer VWDi on the alignment between the stated goal and the empirical evidence), which could be improved with clearer exposition and positioning. A recurring technical concern is that the DISCO objective requires sampling from the denoising posterior $p_{0}(x \mid x_{t})$; this is emphasized by Reviewer Q8EJ and also noted by Reviewers oy6K and Qg4t, and it merits a more explicit tractability discussion and additional numerical evidence (beyond the current Appendix B treatment). Moreover, Reviewer oy6K points out a potential confound in the evaluation pipeline that should be examined in more detail.

Overall, I do not think the rebuttal fully resolves these issues, and I expect the reviewers' ratings would largely remain unchanged.

---

### Decision · Program_Chairs · 2026-01-26

Reject